# GRADIENT-BASED DIVERSITY OPTIMIZATION WITH DIFFERENTIABLE TOP-$k$ OBJECTIVE

**Tianyi Zhou**[1]   **Sebastian Dalleiger**[1]   **Ece Calikus**[2]   **Aristides Gionis**[1]
[1]KTH Royal Institute of Technology, Stockholm, Sweden
[2]Uppsala University, Uppsala, Sweden
`{tzho, sdall, argioni}@kth.se`   `ece.calikus@it.uu.se`

## ABSTRACT

Predicting relevance is a pervasive problem across digital platforms, covering social media, entertainment, and commerce. However, when optimized solely for relevance and engagement, many machine-learning models amplify data biases and produce homogeneous outputs, reinforcing filter bubbles and content uniformity. To address this issue, we introduce a pairwise top-$k$ diversity objective with a differentiable smooth-ranking approximation, providing a model-agnostic way to incorporate diversity optimization directly into standard gradient-based learning. Building on this objective, we cast relevance and diversity as a joint optimization problem, we analyze the resulting gradient trade-offs, and propose two complementary strategies: direct optimization, which modifies the learning objective, and indirect optimization, which reweights training data. Both strategies can be applied either when training models from scratch or when fine-tuning existing relevance-optimized models. We use recommendation as a natural evaluation setting where scalability and diversity are critical, and show through extensive experiments that our methods consistently improve diversity with negligible accuracy loss. Notably, fine-tuning with our objective is especially efficient, requiring only a few gradient steps to encode diversity at scale.

## 1 INTRODUCTION

Modern prediction models are typically evaluated by their ability to maximize accuracy, that is, to rank or classify items in line with ground-truth labels. However, optimizing exclusively for accuracy often yields homogeneous outputs: models repeatedly select similar items, overlook long-tail instances, and fail to provide novel or diverse options. This challenge is particularly evident in recommender systems, which influence decisions in shopping, entertainment, and news (Ricci et al., 2010). However, it also arises broadly in ranking and selection tasks across machine learning (Wang et al., 2023). When diversity is neglected, models risk reinforcing bias, amplifying popularity effects, and reducing the utility of top-$k$ prediction sets (Nguyen et al., 2014; Lambrecht & Tucker, 2019).

Diversity has thus emerged as an important complementary criterion. Recent user studies confirm that diverse outputs can improve satisfaction and engagement when achieved without major loss in relevance (Chen et al., 2018; Holtz et al., 2020; Anderson et al., 2020). The trade-off between relevance and diversity is mainly tackled by either *post-hoc re-ranking methods* or *learning-based methods*. *Post-hoc re-ranking methods* (e.g., MMR, DPP) modify top-$k$ sets to improve diversity (Carbonell & Goldstein, 1998; Chen et al., 2018), but typically suffer from degrading accuracy when diversity increases (Chen et al., 2017). *Model-specific learning-based methods* integrate diversity into training objectives (Borodin et al., 2017; Hurley, 2013; Wang et al., 2023), achieving strong gains with modest accuracy loss. However, these methods rely on opaque models that obscure the source of diversity gains, exhibit slow convergence, and are sensitive to the choice of trade-off parameter balancing relevance and diversity. Additionally, *data-centric approaches* such as augmentation, reweighting, and debiasing (Wang et al., 2021; Lai et al., 2023; Ren et al., 2018; Rastegarpanah et al., 2019) address bias in data distributions, but do not explicitly target diversity. Despite these efforts, we still lack a *unified, differentiable, and model-agnostic framework* for optimizing both relevance and diversity directly during training.

To address these challenges, we propose a unified framework that leverages differentiable ranking to optimize diversity in top-$k$ prediction sets in a scalable and model-agnostic way. At the core, we use an effective diversity objective that can be integrated into the gradient-based training without requiring architecture changes or post-processing. Building on this objective, we introduce two diversification methods. (i) *direct diversity-guided tuning* (DDT), which augments the loss with a joint relevance–diversity term, and (ii) *meta-diversity reweighting* (MDR), which preserves relevance-only training while reweighting data points using the joint loss as a meta-objective. Our approach offers a flexible alternative to post-hoc or model-specific diversification without compromising efficiency.

Our contributions are threefold: (1) We propose a unified differentiable framework for optimizing relevance and diversity in top-$k$ prediction sets, applicable to both end-to-end training and fine-tuning; (2) we provide a theoretical analysis of gradient conflicts, deriving feasible intervals for the trade-off parameter $\beta$ and showing that an adaptive update coincides with the two-objective solution of multi-gradient descent algorithm (MGDA), guaranteeing convergence to Pareto-stationary points; (3) we empirically validate the framework on five benchmark datasets and two model architectures, demonstrating that DDT and MDR achieve substantial diversity improvements with minimal relevance loss, outperforming strong baselines. Notably, the diversity gains extend beyond the explicitly optimized top-$k$ range, reshaping subsequent predictions as well.

## 2 RELATED WORK

Our work is related to diverse recommender systems and multi-objective learning.

**Diversity in recommender systems.** Among the vast literature on recommender systems, the closest are post-hoc and learning-based diversification methods (Zhao et al., 2025); see the survey for a broader overview. *Post-hoc* methods re-rank the output of a relevance-only model to balance relevance and diversity. Representative approaches include maximal marginal relevance (MMR) (Carbonell & Goldstein, 1998), diversity-weighted utility maximization (DUM) (Ashkan et al., 2015), and determinantal point processes (DPP) (Chen et al., 2018). These methods are model-agnostic and easy to implement, but their performance is limited by the quality of the initial relevance ranking, and diversity gain usually comes at a cost of reduced accuracy (Chen et al., 2017).

*Learning-based* approaches incorporate diversity objectives directly into training, including penalties for similarity among recommended items (Hurley, 2013; Wasilewski & Hurley, 2016), formulations that optimize relevance–diversity trade-offs (Wang et al., 2023) list-wise, and graph-based models that encourage coverage of item categories or long-tail exposure (Zheng et al., 2021; Yang et al., 2023). While they often outperform post-hoc re-ranking, they require architectural modifications or adversarial training, making them model-specific and computationally heavy. In contrast, our framework is differentiable and model-agnostic: it can be integrated into standard training pipelines without altering architectures or adding inference overhead.

**Multi-objective learning.** Related is the study of multi-objective optimization for balancing goals such as accuracy, fairness, and revenue (Zheng & Wang, 2022). Classical approaches include scalarization (Paul et al., 2022; Di Noia et al., 2017), which reduces multiple objectives to a single weighted loss, and population-based heuristics such as evolutionary algorithms (Cai et al., 2020), which approximate the Pareto front. While effective in some cases, these approaches either rely on carefully tuned weights or suffer from high computational cost.

More recently, gradient-based methods such as the multi-gradient descent algorithm (MGDA) (Désidéri, 2012) have been applied to recommendation. For instance, MGDA has been used to balance accuracy with fairness (Du et al., 2025; Wu et al., 2022) and with revenue (Milojkovic et al., 2019). These methods guarantee convergence to Pareto-stationary solutions, but their application has so far been limited to objectives other than diversity.

## 3 PROBLEM FORMULATION

In this section, we introduce top-$k$ diversity and our objectives, starting with notation. Given a set of candidate items $\mathcal{I} = \{i_1, i_2, \ldots, i_m\}$ and a collection of users $\mathcal{U} = \{u_1, u_2, \ldots, u_n\}$, the goal is to identify those items in $\mathcal{I}$ that are most relevant for each $u \in \mathcal{U}$. We assume access to a partially-observed supervision matrix $\mathbf{R} \in \mathbb{R}^{n \times m}$, where entries $\mathbf{R}_{u,i}$ represent relevance scores

(e.g., rating, label, or interaction). The set of all *observed* scores is $\Omega = \{(u, i, \mathbf{R}_{u,i})\}^N$, where $N \ll nm$ is the number of observations. We use a relevance prediction model $\mathcal{F}_{\boldsymbol{\Theta}}$ with parameters $\boldsymbol{\Theta}$ to estimate the remaining scores $\tilde{\mathbf{R}}_{u,i} = \mathcal{F}_{\boldsymbol{\Theta}}(u, i)$. To learn such a model, we consider two widely-used approaches. First, we consider *matrix factorization* (MF) (Koren et al., 2009) which predicts ratings $\tilde{\mathbf{R}}_{u,i} = \mathbf{x}_u^\top \mathbf{y}_i$, for user $u$ and item $i$ using the embeddings $\mathbf{x}_u, \mathbf{y}_i \in \mathbb{R}^d$. Second, we consider *neural network models* (He et al., 2017), which predict ratings $\tilde{\mathbf{R}}_{u,i} = \text{MLP}([\mathbf{x}_u; \mathbf{y}_i])$ with a multi-layer perceptron (MLP), allowing non-linear interactions.

In both cases, the models are trained to minimize the regularized *mean squared error* (MSE)

$$\boldsymbol{\Theta}^* = \arg\min_{\boldsymbol{\Theta}} \sum_{(u,i) \in \Omega_\text{T}} (\mathbf{R}_{u,i} - \tilde{\mathbf{R}}_{u,i})^2 + \lambda \|\boldsymbol{\Theta}\|_2^2 \,, \tag{1}$$

between observed relevance scores and predictions. While this approach yields models that predict accurately, neither MF nor MLP is optimized for diversity. Top-$k$ diversity seeks to predict relevance scores $\tilde{\mathbf{r}}_u \in \mathbb{R}^m$ of a user $u$, so that the top-$k$ scores correspond to a set of diverse items $Z_u(k) = \{z_1, \ldots, z_k\}$, indicated by $\mathbf{l}_u = \text{top}_k(\tilde{\mathbf{r}}_u)$. To measure the diversity of the top-$k$ (highest scoring) items, we take a *distance-based* approach (Hassin et al., 1997). In particular, for a given pairwise item-item affinity matrix $\mathbf{S} \in \mathbb{R}^{m \times m}$, we define diversity as the *average pairwise dissimilarity*

$$D_{\mathbf{S}}(Z_u(k)) = \frac{2}{k(k-1)} \sum_{i=1}^m \sum_{j=1}^m \mathbf{l}_u(i)\mathbf{l}_u(j)(1 - \mathbf{S}_{i,j}) \tag{2}$$

of items in $Z_u(k)$. Then, the average top-$k$ diversity is simply

$$\mathcal{L}_{\text{DRO}}(k) = \frac{1}{n} \sum_{u=1}^n D_{\mathbf{S}}(Z_u(k)) \,. \tag{3}$$

While this serves as a natural diversity objective, it cannot be directly optimized with gradients as $\text{top}_k(\cdot)$ involves non-differentiable operations. Our goal, however, is to optimize for relevance and top-$k$ diversity *simultaneously* as part of gradient-based optimization. To achieve this, we relax the non-differentiable diversity reward objective (DRO) using a *differentiable surrogate* (DDRO) next.

**Differentiable diversity.** To overcome the non-differentiability challenge in $\text{top}_k(\cdot)$, we adopt *differentiable ranking* (Blondel et al., 2020), which is a continuous relaxation of sorting. The key idea is to replace the discrete permutation $\mathbf{z}_u$ with a soft ranking vector $\tilde{\mathbf{z}}_u^{(\varepsilon)} \in \mathbb{R}^m$, obtained by projecting the predicted scores $\tilde{\mathbf{r}}_u$ onto the permutahedron $\mathcal{P}_m$—the convex hull of all permutations of $(1, 2, \ldots, m)$ embedded in an $m$-dimensional space. This projection is computed by solving the following entropy-regularized optimization problem (Blondel et al., 2020):

$$\tilde{\mathbf{z}}_u^{(\varepsilon)} = \text{softrank}(\tilde{\mathbf{r}}_u) := \arg\min_{r \in \mathcal{P}_m} \left\{ \frac{1}{\varepsilon} \langle \tilde{\mathbf{r}}_u, r \rangle + H(r) \right\}, \tag{4}$$

where $H(r)$ is the entropy regularizer and $\varepsilon > 0$ controls the approximation smoothness. This definition enables the top-$k$ soft indicator $\tilde{\mathbf{l}}_u(i) = \sigma_\tau(k - \tilde{\mathbf{z}}_u(i))$ using a scaled sigmoid function $\sigma_\tau(x) = [1 + \exp(-x/\tau)]^{-1}$ for user-defined smoothness-sharpness $\tau$, often set to 1 in our experiments. In turn, this makes it possible to train the prediction model end-to-end using diversity-aware gradient updates. Like the discrete counterpart, soft ranking operates with $O(n \log n)$ time and $O(n)$ space complexity. The obvious question is, *does soft ranking lead to sufficiently accurate top-$k$ recommendations?* The better soft ranking approximates the hard ranking, the more reliable are our top-$k$ recommendations, formally summarized in Lemma. 1.

**Lemma 1** (Soft rank approximation (Blondel et al., 2020)). *Given a rating vector $\tilde{\mathbf{r}}_u$, let $\tilde{\mathbf{z}}_u^{(\varepsilon)} \in \mathbb{R}^n$ be the soft rank vector obtained from optimizing* (4). *Then, as $\varepsilon \to 0$, the soft ranks converge to the true ranks of $\tilde{\mathbf{r}}_u$ $\lim_{\varepsilon \to 0} \tilde{\mathbf{z}}_u^{(\varepsilon)} = \text{rank}(\tilde{\mathbf{r}}_u)$ , where $\text{rank}(\tilde{\mathbf{r}}_u) \in \{1, \ldots, n\}^n$ denotes the discrete ranks (breaking ties arbitrarily).*

Replacing indicator in Equation (3) with a the soft-ranking-derived indicator $\tilde{\mathbf{l}}_u$ yields the *differentiable diversity reward objective*

$$\mathcal{L}_{\text{DDRO}} = \frac{1}{n \cdot N} \sum_u^n \sum_{i=1}^m \sum_{j=1}^m \tilde{\mathbf{l}}_u(i)\tilde{\mathbf{l}}_u(j)(1 - \mathbf{S}_{i,j}) \,. \tag{5}$$

To achieve both a high relevance and a high diversity in top-$k$ outputs, we balance both the relevance objective $\mathcal{L}_{\mathrm{rel}}$ and diversity objective $\mathcal{L}_{\mathrm{div}}$ introducing our problem below.

**Problem 2.** *For a given model class $\mathcal{F}_\Theta$, an item affinity matrix $\mathbf{S}$, a user-defined relevance-diversity trade-off $\beta \in [0,1]$; find parameters $\Theta$ that minimize the joint loss*

$$\mathcal{L}_{\mathrm{JOINT}}(\beta, \Theta) = \beta\, \mathcal{L}_{\mathrm{rel}}(\Theta) + (1-\beta)\, \mathcal{L}_{\mathrm{div}}(\Theta)\,. \tag{6}$$

In practice, we take relevance objective $\mathcal{L}_{\mathrm{rel}} = \mathcal{L}_{\mathrm{MSE}}$ and diversity objective $\mathcal{L}_{\mathrm{div}} = -\mathcal{L}_{\mathrm{DDRO}}$. While leading to an efficiently optimizable objective (6), joining them combines two diametrically opposed goals: diversity and relevance. Diverse outputs are not necessarily the most "relevant" ones, and vice versa. We discuss how to deal with this balance in the following section.

Our framework is agnostic to the specific choice of relevance objective. In the main experiments, we adopt a mean-squared error (MSE) loss for concreteness, but any suitable pointwise or ranking loss (e.g., Bayesian Personalized Ranking (BPR)) can be used instead. To demonstrate this flexibility, we report additional results with a BPR-based relevance objective in Appendix E.

## 4  JOINT GRADIENT-BASED TRAINING

In this section, we discuss how this objective can be incorporated into practical training. We propose two strategies: one *direct*, by optimizing the joint loss explicitly, and one *indirect*, by using the joint loss as a meta-objective to reweigh training samples, starting with the direct approach.

### 4.1  BALANCING RELEVANCE AND DIVERSITY

The first approach, called *direct diversity tuning* (DDT), optimize the joint loss in Eq. (6). By utilizing the differentiability of our joint loss, we take an efficient gradient-based optimization approach in which the model parameters are updated using the gradient $\nabla_\Theta \mathcal{L}_{\mathrm{JOINT}}$.

As diversity opposes relevancy, we need to ensure that the optimization converges to a solution that is both relevant and diverse. That is, if both relevance and diversity objectives have gradients that point in similar directions, we say that they are 'aligned'. If the two gradient directions are aligned, any $\beta$ decreases both terms. However, when the gradients $g_{\mathrm{rel}} = \nabla \mathcal{L}_{\mathrm{rel}}$ and $g_{\mathrm{div}} = \nabla \mathcal{L}_{\mathrm{div}}$ are misaligned, any linear combination will necessarily favor one objective at the expense of the other. We want to ensure a descent that ensures a good diversity and accuracy balance for which we realign the gradients. For this, we adaptively compute the optimal balance parameter $\beta^\star$ that ensures simultaneous descent during optimization. Formally, for gradient norms $a = \|g_{\mathrm{rel}}\|$ and $b = \|g_{\mathrm{div}}\|$, we denote the cosine similarity by $\rho = \langle g_{\mathrm{rel}}, g_{\mathrm{div}} \rangle / (ab)$. The combined gradient is $g_\beta = \beta g_{\mathrm{rel}} + (1-\beta)g_{\mathrm{div}}$. For a step along $-g_\beta$ to decrease both objectives simultaneously, the following conditions

$$-\delta_{\mathrm{rel}}(\beta) > 0 \iff \langle g_{\mathrm{rel}}, g_\beta \rangle > 0, \quad -\delta_{\mathrm{div}}(\beta) > 0 \iff \langle g_{\mathrm{div}}, g_\beta \rangle > 0 \tag{A--B}$$

must be satisfied (Désidéri, 2012). Equivalently, the projections of the combined gradient $g_\beta$ onto $g_{\mathrm{rel}}$ and $g_{\mathrm{div}}$ should be positive. This guarantees that a step along $-g_\beta$ decreases both losses at once. We give the feasible region of $\beta$ that satisfies (A-B):

**Lemma 3** (Common descent (Désidéri, 2012)). *For any $a, b > 0$ and $\rho \in [-1, 1]$, the feasible region of $\beta \in [0,1]$ satisfying (A–B) is*

   1. *If $\rho > 0$ (aligned), all $\beta \in [0,1]$ are feasible.*
   2. *If $\rho = 0$ (orthogonal), the feasible set is $\beta \in (0,1)$.*
   3. *If $\rho < 0$ (opposing), the feasible set is $\beta \in \left( \frac{b|\rho|}{a+b|\rho|}, \frac{b}{b+a|\rho|} \right)$.*

*Proof sketch.* The result follows from expanding the directional derivatives $\delta_{\mathrm{rel}}(\beta) = -\beta a^2 - (1-\beta)ab\rho$ and $\delta_{\mathrm{div}}(\beta) = -(1-\beta)b^2 - \beta ab\rho$, and solving the inequalities (A–B) in the three cases $\rho > 0$, $\rho = 0$, and $\rho < 0$. Full details are provided in Appendix A.  □

Even when $\beta$ lies in the feasible interval, different values may still lead to unbalanced overall progress, with one loss improving much more than the other. To avoid this, we select the parameter

$$\beta^\star = \arg \max_{\beta \in [0,1]} \min\{-\delta_{\mathrm{rel}}(\beta),\, -\delta_{\mathrm{div}}(\beta)\}\,,$$

that maximizes the minimum per-step decrease, yielding a *multiple gradient descent algorithm* (MGDA) (Désidéri, 2012) specialized to two objectives. MGDA finds a convex combination of gradients that minimizes the maximum directional derivative across tasks using the closed-form

$$\beta^\star = \frac{b(b - a\rho)}{a^2 + b^2 - 2ab\rho} \ . \tag{7}$$

When $\rho \leq 0$, the ideal balance $\beta^\star$ lies in the feasible interval described in Lemma 3, ensuring valid common descent. Intuitively, $\beta^\star$ equalizes the first-order decreases of relevance and diversity, yielding a balanced update. Finally, when $\beta_t$ is chosen adaptively by the MGDA rule, we obtain convergence to Pareto-stationary solutions.

**Corollary 4** (Pareto-stationarity with adaptive $\beta_t$ (Sener & Koltun, 2018))**.** *Under diminishing step sizes, if each $\beta_t$ is chosen by the minimax rule (or projected variant), then every accumulation point of $\{\Theta^t\}$ is Pareto-stationary for $(\mathcal{L}_{\mathrm{rel}}, \mathcal{L}_{\mathrm{div}})$.*

While the adaptive choice provides balanced progress at each step, a fixed $\beta$ remains useful, as it avoids per-iteration computation. To contextualize our results, we introduce an additional alternative approach which operates with a fixed $\beta$ next. We provide details, incl. pseudocode, in Appendix B.

## 4.2 Optimizing Diversity by Example Reweighting.

A complementary line of work on machine learning fairness and robustness has shown that *reweighting training examples* can effectively mitigate inherent data bias. The central idea is to retain the standard prediction objective, but assign *adaptive weights* to individual samples so that the resulting model better aligns with a criterion (Ren et al., 2018; Rastegarpanah et al., 2019). This approach is naturally formulated as meta-learning: the inner loop minimizes a weighted relevance loss, while the outer loop adjusts weights using a meta-objective that encodes the criterion.

Inspired by this paradigm, we aim to improve diversity by reducing bias at the *data level* through per-sample weights. Intuitively, increasing the importance of 'minor' items increases the chance of their exposure. In brief, the idea is to learn the model parameters by optimizing a weighted relevance loss, while simultaneously learn the weights that balance relevance and diversity. More concretely, our *meta-diversity reweighting* (MDR) introduces a weight $w_{u,i} \in [0, 1]$ for each user-item pair $(u, i)$ in a mini-batch $\mathcal{B}$ and optimizes the *reweighted relevance loss*

$$\mathcal{L}_{\mathrm{MSE}}^w = \sum_{(u,i) \in \mathcal{B}} w_{u,i} \left( \mathbf{R}_{u,i} - \tilde{\mathbf{R}}_{u,i} \right)^2 \ . \tag{8}$$

The algorithm starts with obtaining a temporary model $\Theta'$ (initializing $w_{u,i} = 0$) using a one-step inner update. We then re-evaluate the predictions with the updated parameters and compute the joint meta-loss $\mathcal{L}_{\mathrm{JOINT}}$. Next, we compute the gradient of $\mathcal{L}_{\mathrm{JOINT}}$ with respect to $w$ to obtain a utility score for each sample, and finally normalize the weights of all data points in $\mathcal{B}$ so that they sum to 1. The joint relevance–diversity objective is used only as a *meta-loss* to update the weights $w$, and is never applied directly to model parameters, as detailed in Apx. B.

In contrast to direct diversity-guided tuning, which modifies the training objective itself, MDR preserves the standard relevance-oriented loop while implicitly reshaping the effective data distribution. The meta-objective encourages weights that downplay biased interactions and upweight samples that contribute to both accuracy and diversity. This reweighting perspective allows us to test whether diversity gains can be obtained not only by altering the optimization objective, but also by correcting data imbalance through implicit meta-optimization.

To illustrate the mechanism of MDR, we construct a controlled case study where all items have identical ratings but belong to distinct diversity clusters. After a one-step meta update, the gradient of the joint objective with respect to sample weights assigns larger normalized weights to items that increase inter-cluster coverage, while suppressing redundant items within the same cluster. This example demonstrates that MDR automatically emphasizes samples that contribute most to diversity without explicit cluster supervision (see Appendix C for details).

We evaluate both DDT and MDR in two settings: *from-scratch training*, where relevance and diversity are optimized jointly, and *fine-tuning*, where a relevance-trained model is adapted for diversity.

Table 1: Statistics of datasets and diversity metrics.

| Dataset | $|\mathcal{U}|$ | $|\mathcal{I}|$ | $|\Omega|$ | DRO($\mathcal{I}$) | $|\mathcal{C}|$ |
|---------|------|------|-----------|-----------|-----|
| Coat | 290 | 300 | 6 960 | 0.73 | 33 |
| KuaiRec | 1 411 | 3 327 | 4 676 570 | 0.91 | 31 |
| Netflix | 4 999 | 1 112 | 557 176 | 0.83 | 27 |
| Yahoo-R2 | 4 050 | 5 000 | 684 782 | 0.26 | 58 |
| MovieLens | 6 040 | 3 706 | 1 000 208 | 0.83 | 18 |

## 5 EXPERIMENTAL EVALUATION

We describe the experimental setup to evaluate the effectiveness of our proposed solutions, introducing the datasets and evaluation criteria in this section, and provide further details in Appendix D and E.

**Datasets.** We evaluate our methods across three domains: *entertainment*, *product*, and *social* recommendations. For entertainment, we use MovieLens, Netflix, and Yahoo-R2, which contain user ratings on movies or music with genre/category annotations. For product recommendation we use the Coat dataset, which includes user ratings and product attributes. For social recommendation we use KuaiRec, a large-scale mobile video dataset with watch-time based ratings. The item affinity matrices $\mathbf{S}$ are pre-computed by Jaccard similarity scores based on genre/category information of items. Basic statistics and detailed preprocessing are given in the Appendix D.

**Baselines.** We compare against relevance- and diversity-aware approaches: (i) *Non-negative Matrix Factorization* (NMF) (Lee & Seung, 1999) as a relevance-only baseline; (ii) greedy post-processing methods such as *Maximal Marginal Relevance* (MMR) (Carbonell & Goldstein, 1998) and *Diversity-weighted Utility Maximization* (DUM) (Ashkan et al., 2015); (iii) probabilistic diversification via *Determinantal Point Processes* (DPP) (Chen et al., 2018); and (iv) the graph-based *Diversified GNN Recommender* (DGRec) (Yang et al., 2023). We use the authors' publicly-available implementations.

**Evaluation criteria.** We evaluate accuracy using *hit rate*, *precision*, and *recall*. For each user $u$, let $\mathcal{R}_u$ denote the top-$k$ recommended items and $\mathcal{T}_u$ the set of ground-truth relevant items (i.e., rated above a threshold), the *Hit rate* measures whether at least one relevant item appears in $\mathcal{R}_u$; *precision* is the fraction of items in $\mathcal{R}_u$ that are in $\mathcal{T}_u$; and *recall* is the fraction of relevant items in $\mathcal{T}_u$ that are retrieved in $\mathcal{R}_u$. When ground-truth is unknown, we report the estimated user satisfaction using preference likelihoods $p(\tilde{\mathbf{R}}_{u,i}) \in [0, 1]$ and threshold $\tau = 0.8$ as the *relevance score*

$$\text{Relevance}(u) = \frac{1}{k} \sum_{i \in \mathcal{R}_u} \mathbb{I}[p(\tilde{\mathbf{R}}_{u,i}) > \tau] \, . \tag{9}$$

### 5.1 EXPERIMENTAL ANALYSIS

Having introduced our setup, we now introduce our research questions and experimental analysis.

**Q1** Does adaptive $\beta$ behave as predicted by our theory?
**Q2** How does the choice of $\beta$ affect the accuracy–diversity trade-off?
**Q3** How do our approaches compare with established diversification methods?
**Q4** How do diversity gains evolve with varying $k$?

In the following, we experimentally answer all these questions in detail.

**Evaluating adaptive $\beta$ optimization.** We first study the direct optimization approach using both *from-scratch training* and *fine-tuning* of a relevance-pretrained model. At each step, we adopt the adaptive $\beta$ rule, computing both the optimal coefficient $\beta^\star$ and the feasible interval from Lemma 3. Figure 1a shows from-scratch training and fine-tuning for 100 epochs on MovieLens. We employ the *neural collaborative filtering* (NCF) model with an MLP architecture, and consider top-10 diversity.

In the top row of Figure 1a, we observe that the adaptive coefficient $\beta^\star$ remains consistently within the feasible interval defined by Lemma 3. We see how $\beta^\star$ gradually shifts from favoring relevance toward favoring diversity, confirming that the adaptive rule preserves a common descent direction.

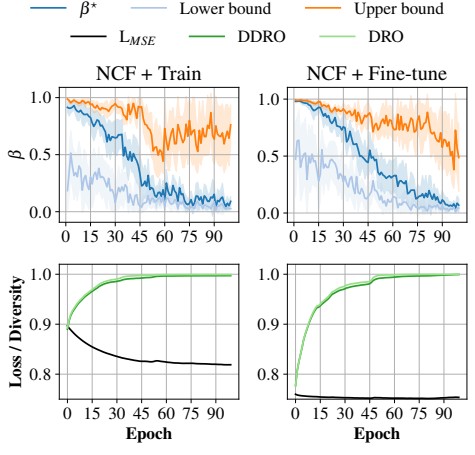 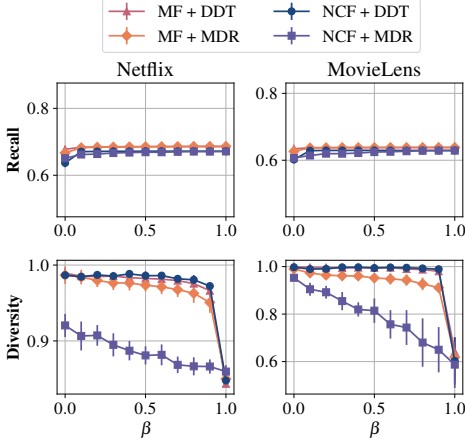

(a) Optimization with adaptive $\beta$ (MovieLens).  (b) Optimization with varying 'static' $\beta$s.

Figure 1: Optimization of relevance-diversity joint objective. Figure 1a shows results of training and fine-tuning of NCF models with adaptive $\beta$ on the MovieLens dataset. Figure 1b shows the performance comparison of DDT and MDR applied to two models (NMF and NCF) across two datasets with varying fixed $\beta \in [0, 1]$.

Table 2: Performance of DDT, MDR, and five competitors in terms of diversity and relevance across Coat, Yahoo-R2, Netflix, MovieLens, and KuaiRec. We highlight the the best results with **bold**, and underline the second best, reporting mean and standard deviation over 10 trials.

| Algorithm | Coat | | Yahoo-R2 | | Netflix | | MovieLens | | KuaiRec | |
|---|---|---|---|---|---|---|---|---|---|---|
| | Diversity | Relevance | Diversity | Relevance | Diversity | Relevance | Diversity | Relevance | Diversity | Relevance |
| NMF | 0.77 (0.02) | 0.41 (0.04) | 0.09 (0.05) | 0.76 (0.02) | 0.84 (0.01) | 0.88 (0.03) | 0.62 (0.08) | 0.98 (0.00) | 0.89 (0.01) | 0.84 (0.01) |
| MMR | 0.80 (0.01) | 0.40 (0.04) | 0.80 (0.06) | 0.68 (0.03) | 0.93 (0.01) | 0.85 (0.04) | 0.94 (0.03) | 0.95 (0.01) | 0.99 (0.00) | 0.76 (0.01) |
| DUM | 0.81 (0.01) | 0.31 (0.04) | 0.98 (0.02) | 0.60 (0.02) | 0.93 (0.00) | 0.71 (0.03) | 0.93 (0.01) | 0.91 (0.01) | 0.98 (0.00) | 0.42 (0.01) |
| DPP | 0.81 (0.01) | 0.39 (0.04) | **1.00 (0.00)** | 0.58 (0.01) | 0.96 (0.00) | 0.84 (0.04) | 0.98 (0.01) | 0.95 (0.01) | **1.00 (0.00)** | 0.75 (0.01) |
| DGRec | 0.71 (0.01) | **0.69 (0.02)** | 0.33 (0.01) | 0.83 (0.02) | 0.76 (0.00) | 0.83 (0.01) | 0.73 (0.01) | 0.47 (0.02) | 0.91 (0.02) | 0.18 (0.04) |
| DDT | **0.83 (0.01)** | 0.50 (0.05) | 0.98 (0.01) | **0.85 (0.01)** | **0.98 (0.00)** | **0.97 (0.01)** | **1.00 (0.00)** | **1.00 (0.00)** | 0.98 (0.02) | **0.95 (0.00)** |
| MDR | 0.82 (0.01) | 0.47 (0.03) | 0.86 (0.09) | 0.82 (0.02) | **0.98 (0.01)** | 0.93 (0.02) | 0.97 (0.02) | 0.99 (0.00) | 0.97 (0.01) | 0.85 (0.01) |

The bottom row reports the optimization trajectories. The relevance loss ($\mathcal{L}_{\text{MSE}}$) decreases steadily while diversity (DRO and DDRO) increases, showing that adaptive $\beta$ achieves a balanced trade-off in both training objectives. We see that fine-tuning preserves the relevance of the pretrained model, whereas from-scratch converges to a worse stationary point.

We further observe that fine-tuning preserves the relevance performance of the pretrained model, whereas training from scratch reduces $\mathcal{L}_{\text{MSE}}$ but converges to a worse stationary point. In contrast, both approaches reach a similar level of diversity, suggesting that regularization may hinder the convergence towards more 'relevant' models during from-scratch optimization. Fine-tuning, on the other hand, starts with a relevance-optimized model, which we effectively tune for diversity. Finally, because DRO and DDRO are close, we empirically validated the accuracy of our relaxation. We discuss the impact of the parameter $\epsilon$ on approximation accuracy later. Overall, we saw that adaptive $\beta$ effectively delivers simultaneous improvements in relevance and diversity.

**Relevance-diversity trade-off.** We next examine how the choice of static $\beta$ affects relevance and diversity. Figure 1b reports recall and diversity for DDT and MDR with MF and NCF on Netflix and MovieLens. Starting from a relevance-trained model, we fine-tune with the joint objective (6) while varying $\beta \in [0, 1]$, ranging from $\beta = 1$ (pure relevance) to $\beta = 0$ (pure diversity). We run 10 random trials for each experiment and report the mean and variance. We observe a stable low-variance recall across a wide range of $\beta$, showing that introducing diversity does not substantially compromise accuracy with fixed $\beta$. A noticeable drop occurs only when $\beta$ approaches zero, where the objective focuses almost exclusively on diversity. In contrast, diversity improves as $\beta$ decreases. Moreover, DDT consistently outperforms MDR in terms of diversity with a neural network model, while the two strategies behave more similarly with a matrix factorization model, suggesting that

reweighting is particularly beneficial for simpler low-rank models. These findings demonstrate that fixed-$\beta$ optimization offers a practical means to explore the relevance–diversity trade-off, enabling substantial diversity gains with minimal loss of accuracy. Similar patterns are observed for MSE loss, hit rate, and precision, with detailed results on additional datasets provided in Appendix E.

Table 3: Diversity gain achieved by Direct Diversity Tuning (DDT) on top-$1 \sim k$ and $k + 1 \sim 2k$ recommendations across five datasets Coat, Yahoo-R2, Netflix, MovieLens, and KuaiRec. We vary the diversity reward parameter $k \in \{5, 10, 20, 30, 40\}$ and apply DDT to pre-trained NMF and NCF, reporting diversity gain. We report the mean and standard deviation across 10 runs.

| | Dataset | Diversity gain (%) | | | | | | | | | |
|---|---|---|---|---|---|---|---|---|---|---|---|
| | | $1 \sim k$ | | | | | $k+1 \sim 2k$ | | | | |
| | | $k{=}5$ | $k{=}10$ | $k{=}20$ | $k{=}30$ | $k{=}40$ | $k{=}5$ | $k{=}10$ | $k{=}20$ | $k{=}30$ | $k{=}40$ |
| **NMF** | Coat | 10.9 (4.3) | 5.4 (1.8) | 3.5 (1.7) | 2.9 (1.1) | 2.6 (0.8) | -0.5 (4.7) | -2.5 (4.4) | -1.8 (1.3) | -1.5 (1.7) | -1.9 (0.8) |
| | Yahoo-R2 | 75.0 (14.3) | 77.4 (8.7) | 78.5 (3.8) | 78.4 (2.4) | 77.2 (1.6) | 54.9 (11.2) | 50.8 (9.9) | 47.1 (2.7) | 45.2 (2.1) | 41.9 (1.3) |
| | Netflix | 14.7 (2.3) | 13.8 (1.0) | 11.3 (0.7) | 10.2 (0.6) | 9.6 (0.5) | 7.8 (2.6) | 4.2 (1.7) | 3.1 (2.1) | 2.4 (2.2) | 2.2 (1.8) |
| | MovieLens | 41.4 (13.1) | 37.2 (8.3) | 28.4 (4.6) | 23.8 (3.2) | 21.6 (2.6) | 25.6 (10.4) | 16.2 (5.1) | 10.1 (2.8) | 8.3 (2.3) | 6.8 (1.6) |
| | KuaiRec | 9.2 (1.7) | 7.2 (0.5) | 9.4 (0.8) | 10.6 (0.7) | 10.9 (0.8) | 5.5 (1.5) | 10.3 (2.1) | 6.0 (2.0) | 3.3 (1.0) | 1.4 (1.4) |
| **NCF** | Coat | 9.7 (2.4) | 1.2 (1.8) | 6.9 (1.3) | 0.3 (0.8) | 4.8 (0.5) | -0.9 (0.6) | 3.5 (0.4) | -1.1 (0.7) | 2.7 (0.5) | -1.0 (0.5) |
| | Yahoo-R2 | 79.5 (12.8) | 9.4 (17.5) | 79.3 (7.6) | 4.6 (6.0) | 78.2 (8.7) | 4.9 (13.7) | 77.5 (4.6) | 0.6 (9.0) | 77.0 (6.3) | 1.7 (7.3) |
| | Netflix | 13.7 (2.0) | 8.2 (1.9) | 13.3 (0.9) | 4.5 (1.4) | 10.8 (0.7) | 3.9 (1.2) | 9.5 (0.5) | 3.6 (1.1) | 9.0 (0.5) | 3.2 (1.1) |
| | MovieLens | 46.8 (13.0) | 19.6 (6.6) | 36.2 (7.8) | 12.7 (3.0) | 26.9 (4.5) | 10.9 (1.9) | 22.9 (3.0) | 9.6 (1.5) | 20.9 (2.5) | 9.0 (1.5) |
| | KuaiRec | 8.8 (1.3) | 5.4 (1.8) | 6.5 (0.3) | 10.7 (2.6) | 8.7 (0.7) | 8.0 (1.2) | 10.6 (0.7) | 4.2 (0.9) | 11.1 (0.5) | 2.3 (0.9) |

**Diversity-relevance performance of all approaches.** Next, we compare the proposed methods against alternative diversification approaches across all datasets. Fixing $k = 10$ and $\beta = 0.2$, we fine-tune a pre-trained relevance-optimized matrix factorization model for 100 epochs and select the most diverse checkpoint. In Table 2 we report the mean and standard deviation of diversity and relevance scores for all methods from 10 trials.

We see that our direct diversity tuning approach (DDT) demonstrates strong performance in both relevance and diversity, achieving either the best or second-best results across nearly all metrics and datasets, showing the effectiveness of jointly optimizing. Similarly, we see that the implicit data reweight approach (MDR) shows highly competitive performance. Its performance is particularly notable on Netflix, MovieLens and KuaiRec, where it approaches or matches the performance of DDT. This supports our earlier observation that implicit, data-driven reweighting can offer strong benefits. On the other hand, while post-hoc diversification of NMF using MMR, DUM, and DPP, considerably improve top-$k$ diversity, they often notably reduce relevance, aligning with the results reported in previous studies (Chen et al., 2017). For example, DPP occasionally achieves the highest diversity—particularly on Yahoo-R2 and KuaiRec—but at a cost of low relevance. Greedy methods like MMR and DUM yield moderate diversity improvements but underperform in relevance. DGRec excels in diversity on certain datasets but suffers from severe relevance degradation, especially on KuaiRec, with a diversity of 0.91 and a relevance of 0.18. In contrast, DDT and MDR maintain a significantly better trade-off. In summary, fine-tuning with direct and implicit methods both achieve superior relevance-diversity trade-offs across the board, outperforming the competitors.

**Diversity gain with varying $k$.** We examine how diversity changes across the growing number of recommended items. Starting with relevance-optimized matrix factorization and neural network models, we fine-tune with the joint objective $\mathcal{L}_{\text{JOINT}}$ with DDT for 10 epochs at a fixed $\beta = 0.5$, and vary $k \in \{5, 10, 20, 30, 40\}$, selecting the best results. Since larger top-$k$ sets approach coverage of the entire item space, it becomes impossible to improve diversity beyond the dataset's average. To account for this, we report *relative diversity gains* over the relevance-only baseline, normalized by the maximum achievable score. Specifically, we measure DRO gains as the difference between fine-tuned and pre-trained models, normalized to lie in $[0, 1]$ and reported as percentages. We distinguish between *in-objective* gains (top-$k$) and *out-of-objective* gains (the subsequent $k+1 \sim 2k$ items).

In Table 3, we observe positive *in-objective* diversity gains across all datasets. In Coat, Netflix, and MovieLens, we see a decreasing gain as $k$ increases. The improvement is especially pronounced on Yahoo-R2, with diversity gains exceeding $78.5 \pm 3.8\%$ in NMF and $79.3 \pm 7.6$ in NCF at $k = 20$, due to its low initial diversity (e.g., 0.09 in the base model). Substantial gains are also observed on MovieLens (e.g., $41.4 \pm 13.1\%$ and $46.8 \pm 13\%$ at $k = 5$) and Netflix, demonstrating the effectiveness of our method across both sparse and dense recommendation scenarios.

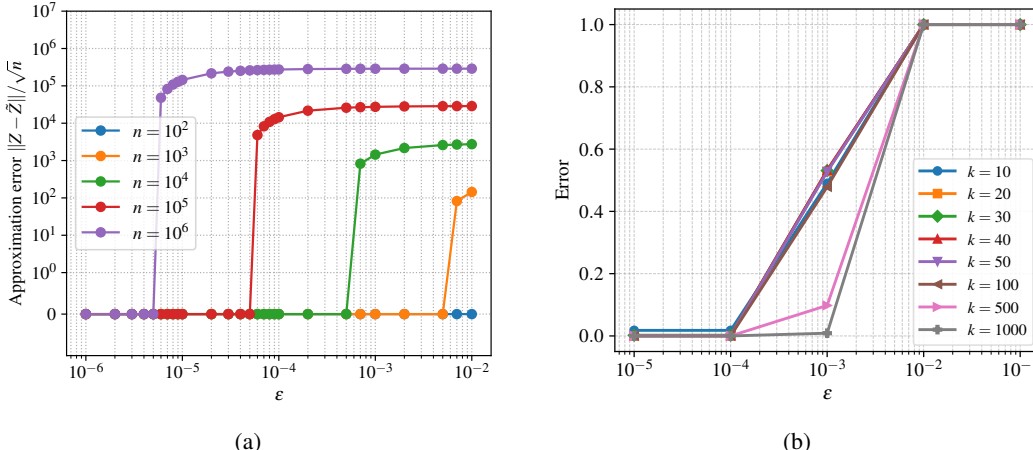

Figure 2: Ablation study on approximation errors for ranking and diversity. (a) Rank approximation error with varying parameter $\varepsilon$ and item size. The $y$-axis reports the approximation error between the exact rank and the softrank approximation, $\|\mathbf{Z} - \tilde{\mathbf{Z}}\|_2/\sqrt{n}$, and the $x$-axis reports the parameter $\varepsilon$. Setting $\varepsilon = 1/n$ yields zero approximation error, and there is a tunable range of $\varepsilon$ that achieves small error while allowing larger values. (b) Diversity approximation error of DDRO with different $\varepsilon$ and $k$ on MovieLens. The $y$-axis reports the approximation error $|\mathcal{L}_{\mathrm{DRO}} - \mathcal{L}_{\mathrm{DDRO}}|/\mathcal{L}_{\mathrm{DRO}}$.

**Generality beyond the optimized range.** We next ask whether the benefits of diversity optimization extend beyond the explicitly encoded objective. To this end, we evaluate diversity not only at the target top-$k$ set but also in the subsequent $k+1 \sim 2k$ items in Table 3. We observe broadly positive *out-of-objective* gains, which again diminish as $k$ increases. This effect is especially strong on Yahoo-R2 and MovieLens, showing that optimization reshapes the ranking itself: improvements are not limited to the items directly optimized, but generalize to deeper parts of the recommendation list. In contrast, results on Coat fluctuate around zero, likely due to its small scale and limited item pool. These findings highlight the generality of our diversity-guided optimization: it improves diversity not only within the objective's explicit target (top-$k$) but also beyond, demonstrating that the learned updates capture a broader notion of diversity than what is directly optimized.

**Approximation error with varying $\epsilon$.** In Figure 2a, we first examine the rank approximation error induced by the softrank operator. We report the discrepancy between the exact rank and its differentiable approximation, measured as $\|\mathbf{Z} - \tilde{\mathbf{Z}}\|_2/\sqrt{n}$, as a function of $\epsilon$ and the number of items. Consistent with the theoretical construction, setting $\epsilon = 1/n$ yields essentially zero approximation error. Moreover, there exists a range of larger $\epsilon$ values that still achieve small error, indicating that one can trade off a modest loss in approximation accuracy for improved numerical stability and smoother gradients. In Figure 2b, we report the approximation error between the exact diversity DRO and its differentiable surrogate DDRO as the top-$k$ size varies. We define the error as $|\mathcal{L}_{\mathrm{DRO}} - \mathcal{L}_{\mathrm{DDRO}}|/\mathcal{L}_{\mathrm{DRO}}$. For small $k$ (e.g., $k < 100$), setting $\epsilon < 10^{-4}$ already yields a close approximation. As $k$ increases, however, a smaller $\epsilon$ is required to maintain the same accuracy. This highlights that the choice of $\epsilon$ is critical: an improper value can lead to arbitrarily large errors, rendering the diversity measure unreliable for optimization.

**Case study.** To demonstrate the utility of our approaches, we examine the recommendations generated from an NMF model and its diversity-optimized variant. In Table 4, we observe that NMF produces homogeneous items where six out of ten items are labeled as *drama*, with remaining entries only marginally extending into *romance*, *documentary*, *comedy* or *War*. In contrast, the DDT-generated list spans a much broader range of genres, including *horror*, *thriller*, *animation*, *crime*, and *adventure*. Despite diversity gains, DDT retains three of the top items in relevance-optimized recommendation (*Mamma Roma*, *Smashing Time*, and *Gate of Heavenly Peace*), which collectively represent the relevant genre themes (*drama*, *comedy*, and *documentary*, respectively), suggesting that DDT successfully preserves relevant while enhancing diversity.

Table 4: Comparison of recommendation lists for a user from DDT and NMF. Movies appearing in both lists are **bold**.

| | DDT | | | NMF | |
|---|---|---|---|---|---|
| Rank | Genres | Movie Title | Rank | Genres | Movie Title |
| 1 | Horror | Vampyros Lesbos (Las Vampiras) | 1 | Drama | **Mamma Roma** |
| 2 | Thriller | The Spiral Staircase | 2 | Drama | Foreign Student |
| 3 | War | Prisoner of the Mountains | 3 | Drama | The Apple |
| 4 | Animation, Musical | Melody Time | 4 | Drama, Romance | Leather Jacket Love Story |
| 5 | Documentary | **The Gate of Heavenly Peace** | 5 | Comedy | **Smashing Time** |
| 6 | Crime | Lured | 6 | Documentary | **The Gate of Heavenly Peace** |
| 7 | Drama | **Mamma Roma** | 7 | Documentary | Modulations |
| 8 | Comedy | **Smashing Time** | 8 | Comedy, Romance, War | Forrest Gump |
| 9 | Adventure | Ulysses (Ulisse) | 9 | Drama | Schlafes Bruder (Brother of Sleep) |
| 10 | Romance | Persuasion | 10 | Drama, War | Schindler's List |

## 6 CONCLUSION

We addressed the issue of limited top-$k$ diversity of relevance prediction models, which contributes to echo chambers, reduced novelty, and social polarization. By integrating diversity into a gradient-based optimization, we presented a unified framework for diversity-aware recommendation by introducing a differentiable diversity objective that enables end-to-end optimization of both relevance and diversity. We proposed two complementary, model-agnostic algorithms to support explicit and implicit integration of diversity into standard recommender systems. With extensive experiments on real-world datasets we demonstrated that our methods consistently improve diversity, converge efficiently, and introduce minimal computational overhead.

**Limitations and future work.** While our framework effectively promotes both relevance and diversity in top-$k$ recommendations, several challenges remain. First, the adaptive $\beta$ is applicable only for a direct joint objective optimization, while it is optimal in each step, but the optimization trajectory may converge to a compromised stationary point, which is not globally optimal. The mechanism that leads to a global optimal solution remains unexplored. Future work includes developing new ways to balance our competing goals via multi-objective formulations. Second, we primarily relies on categorical similarity to quantify diversity. While this provides interpretability, it does not capture more nuanced relationships or learned semantic embeddings. Future work extends the research into more expressive and context-sensitive diversity metrics. Finally, as diversity bias stems from the training data—implicit and explicit approaches often achieve similar performance levels—we see significant potential in the data-driven diversification, such as counterfactual data augmentation or diversity-aware sampling.

Despite these research opportunities, we observe that our methods significantly increase diversity with only an imperceptible decrease in relevance.

## ETHICS STATEMENT

This work is primarily theoretical and focuses on incorporating diversity into machine-learning models for ranking. All datasets used in our experiments are publicly available, and no personally identifiable or sensitive information was collected or processed. No user studies or interventions involving human participants were conducted.

While our approach is motivated by the goal of improving diversity in ranking outcomes, we acknowledge that any deployment of such methods may have broader societal implications. Potential concerns include including fairness considerations, potential biases in the underlying data, or unintended effects depending on the application context. We leave a deeper exploration of these implications to future research.

## REPRODUCIBILITY STATEMENT

Our code is available in an online repository.[1].

## AI USAGE DISCLOSURE

Large language models (LLMs) were employed during the research phase to assist with surveying related literature, including identifying and summarizing relevant papers and methods, as well as sketching and testing proposed approaches described in prior work. Generative models were also used to draft scripts for data processing (e.g., formatting and visualizing data, preliminary experimental validation). These scripts were only used for exploration and are not part of the final experimental pipeline or released codebase. In addition, ChatGPT and Grammarly were used to assist with grammar and phrasing in the manuscript. All outputs from these tools were reviewed, edited, and verified by the authors, who take full responsibility for the final content.

## ACKNOWLEDGEMENTS

We thank all anonymous reviewers and the area chair for their valuable feedback and constructive suggestions. This research was supported by the ERC Advanced Grant REBOUND [834862], the Swedish Research Council (VR) [2024-05603], the European Commission MSCA DN [101168951], and the Wallenberg AI, Autonomous Systems and Software Program (WASP) funded by the Knut and Alice Wallenberg Foundation.

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

## A  PROOFS

In this appendix, we provide detailed proofs for the results on the trade-off parameter $\beta$.

### A.1  PROOF OF LEMMA 3

In the joint loss, we minimize both

$$\mathcal{L}_{\text{rel}} := \mathcal{L}_{\text{MSE}}, \qquad \mathcal{L}_{\text{div}} := -\mathcal{L}_{\text{DDRO}},$$

so decreasing $\mathcal{L}_{\text{div}}$ increases diversity.

We denote the gradients

$$g_{\text{rel}} = \nabla \mathcal{L}_{\text{rel}}, \qquad g_{\text{div}} = \nabla \mathcal{L}_{\text{div}}, \qquad g_\beta = \beta g_{\text{rel}} + (1-\beta)g_{\text{div}}, \quad \beta \in [0,1],$$

so that direct diversity-guided tuning (DDT) performs

$$\Theta^+ = \Theta - \eta g_\beta.$$

Let

$$a = \|g_{\text{rel}}\| > 0, \qquad b = \|g_{\text{div}}\| > 0, \qquad \rho = \frac{\langle g_{\text{rel}}, g_{\text{div}}\rangle}{ab} \in [-1,1].$$

For an infinitesimal step along $-g_\beta$, the first-order changes in each objective are

$$\delta_{\text{rel}} = \frac{d}{dt}\mathcal{L}_{\text{rel}}(\Theta - tg_\beta)\Big|_{t=0} = -\langle g_{\text{rel}}, g_\beta\rangle,$$

$$\delta_{\text{div}} = \frac{d}{dt}\mathcal{L}_{\text{div}}(\Theta - tg_\beta)\Big|_{t=0} = -\langle g_{\text{div}}, g_\beta\rangle.$$

Substituting $g_\beta = \beta g_{\text{rel}} + (1-\beta)g_{\text{div}}$ gives

$$\delta_{\text{rel}} = -\beta\|g_{\text{rel}}\|^2 - (1-\beta)\langle g_{\text{rel}}, g_{\text{div}}\rangle = -\beta a^2 - (1-\beta)ab\rho,$$

$$\delta_{\text{div}} = -(1-\beta)\|g_{\text{div}}\|^2 - \beta\langle g_{\text{div}}, g_{\text{rel}}\rangle = -(1-\beta)b^2 - \beta ab\rho.$$

We require both decreases:

$$\delta_{\text{rel}} < 0, \qquad \delta_{\text{div}} < 0.$$

Rearranging each inequality yields the following necessary and sufficient conditions:

$$\beta a + (1-\beta)b\rho > 0, \tag{A}$$

$$(1-\beta)b + \beta a\rho > 0. \tag{B}$$

We now analyze (A) and (B) systematically.

*Inequality (A).*

$$\beta a + (1-\beta)b\rho > 0 \iff \beta(a - b\rho) > -b\rho.$$

- If $a - b\rho > 0$, then $\beta > \frac{-b\rho}{a-b\rho}$, we need further analyze $\rho$.
- If $a - b\rho < 0$, then $\beta < \frac{-b\rho}{a-b\rho}$, we need further analyze $\rho$.
- If $a - b\rho = 0$, then the inequality holds for all $\beta$.

*Inequality (B).*

$$(1-\beta)b + \beta a\rho > 0 \iff b + \beta(a\rho - b) > 0.$$

- If $a\rho - b > 0$, then $\beta > \frac{-b}{a\rho-b}$ (RHS negative, vacuous for $\beta \in [0,1]$).
- If $a\rho - b < 0$, then $\beta < \frac{b}{b-a\rho}$, we need further analyze $\rho$.
- If $a\rho = b$, then the inequality holds for all $\beta$.

We specialize the above formulas to the three relevant regimes of $\rho$.

*Case I: Aligned gradients ($\rho > 0$).* Here the two gradients point in similar directions. We revisit inequalities (A) and (B).

*Condition (A).*

$$\beta a + (1 - \beta)b\rho > 0 \iff \beta(a - b\rho) > -b\rho.$$

- If $a - b\rho \geq 0$, then the right-hand side is nonpositive ($-b\rho \leq 0$ since $\rho > 0$). Thus the inequality is automatically satisfied for all $\beta \in [0, 1]$; no restriction is imposed.

- If $a - b\rho < 0$, then the inequality gives $\beta < \frac{-b\rho}{a - b\rho}$. But since $a - b\rho < 0$ and $-b\rho < 0$, the right-hand side is positive. In fact, one can check that $\frac{-b\rho}{a - b\rho} > 1$. Therefore any $\beta \in [0, 1]$ still satisfies the inequality.

*Condition (B).*

$$(1 - \beta)b + \beta a\rho > 0 \iff b + \beta(a\rho - b) > 0.$$

- If $a\rho - b \geq 0$, then the right-hand side is increasing in $\beta$ and at $\beta = 0$ equals $b > 0$, so the inequality holds for all $\beta \in [0, 1]$.

- If $a\rho - b < 0$, then the inequality becomes $\beta < \frac{b}{b - a\rho}$. Since $a\rho - b < 0$, the denominator $b - a\rho > 0$ and thus the right-hand side exceeds 1. Hence the condition imposes no restriction within $\beta \in [0, 1]$.

Both inequalities (A) and (B) are therefore automatically satisfied when $\rho > 0$. Thus

$$\delta_{\text{rel}} < 0 \quad \text{and} \quad \delta_{\text{div}} < 0 \qquad \text{for every } \beta \in [0, 1].$$

In other words, when gradients are aligned, any convex combination yields a valid common descent direction. At $\beta = 0$ or $\beta = 1$, one loss still strictly decreases while the other is nonincreasing; for $0 < \beta < 1$, both decrease strictly.

*Case II: Orthogonal gradients ($\rho = 0$).* Plugging $\rho = 0$ into (A) and (B) gives

$$\beta a > 0 \iff \beta > 0, \qquad (1 - \beta)b > 0 \iff \beta < 1.$$

$$\implies \delta_{\text{rel}} < 0, \ \delta_{\text{div}} < 0 \quad \text{for all } \beta \in (0, 1).$$

*Case III: Opposing gradients ($\rho < 0$).* Let $\rho = -|\rho| < 0$. Then

$$\text{From (A): } \beta > \frac{b|\rho|}{a + b|\rho|} =: L, \qquad \text{From (B): } \beta < \frac{b}{b + a|\rho|} =: U.$$

Thus the feasible interval is

$$\beta \in (L, U) = \left( \frac{b|\rho|}{a + b|\rho|}, \ \frac{b}{b + a|\rho|} \right).$$

This interval is nonempty whenever $|\rho| < 1$, since

$$U - L = \frac{ab(1 - |\rho|^2)}{(b + a|\rho|)(a + b|\rho|)} > 0.$$

If $|\rho| = 1$ (exactly opposite gradients), then any $\beta \in (0, 1)$ still makes both $\delta_{\text{rel}}, \delta_{\text{div}} < 0$.

We have characterized the feasible region of $\beta$:

- If $\rho > 0$: every $\beta \in [0, 1]$ yields common descent.
- If $\rho = 0$: every $\beta \in (0, 1)$ yields common descent.
- If $\rho < 0$: feasible region is $(L, U)$ as defined above.

This completes the proof of Lemma 3. $\qquad\square$

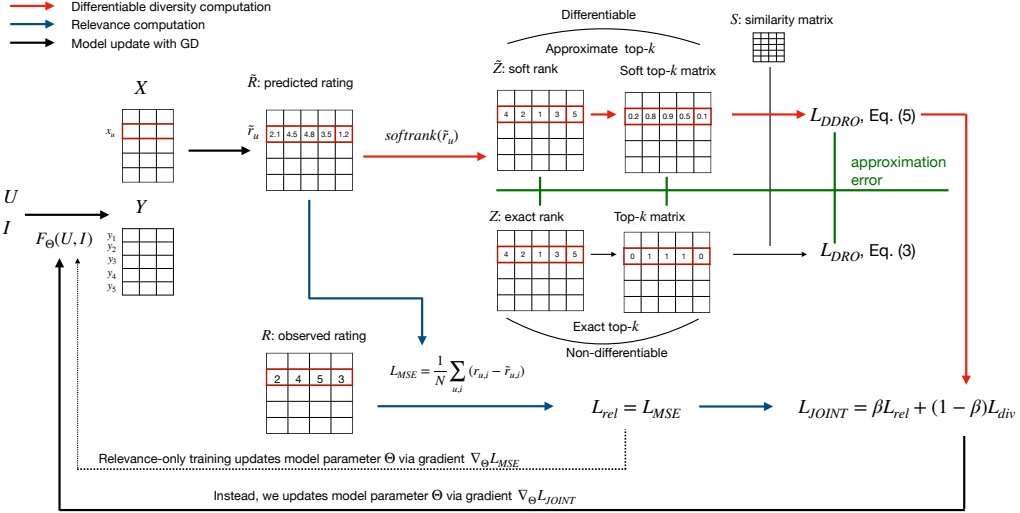

Figure 3: DDT workflow

## A.2 Proof of $\beta^{\star}$

We seek $\beta^{\star}$ such that $-\delta_{\mathrm{rel}}(\beta) = -\delta_{\mathrm{div}}(\beta)$. Expanding:

$$\beta a^2 + (1 - \beta)ab\rho = (1 - \beta)b^2 + \beta ab\rho.$$

Rearranging,

$$(a^2 + b^2 - 2ab\rho)\beta = b^2 - ab\rho.$$

Thus

$$\beta^{\star} = \frac{b(b - a\rho)}{a^2 + b^2 - 2ab\rho}.$$

When $\rho \leq 0$, substitution verifies $L < \beta^{\star} < U$, where $(L, U)$ is the interval from Lemma 3. Hence $\beta^{\star}$ is feasible and maximizes the minimum first-order decrease.

$\square$

## B Algorithms

**Algorithm workflow diagram** Figure 3 and 4 illustrate the workflow of DDT and MDR.

DDT directly optimizes the joint loss $\mathcal{L}_{\mathrm{JOINT}} = \beta\mathcal{L}_{\mathrm{MSE}} + (1 - \beta)\mathcal{L}_{\mathrm{DDRO}}$ via standard gradient descent using innovative soft-ranking, top-$k$ selection, pairwise diversity penalties, and adaptive diversity weights.

For MDR, in a nutshell, the meta learning procedure reweights using two different stages: the model stage and the design stage. While the design stage updates $w$ using gradients $\nabla_w \mathcal{L}_{\mathrm{JOINT}}$ from loss + diversity; the model stage updates the model $\Theta$ through gradients $\nabla_\Theta \mathcal{L}_{MSE}^w$ of the $w$-weighted relevance objective. In detail, we consider the outer loop to be the standard mini-batch training procedure that updates the model parameters for each mini-batch of data. For each data point in the mini-batch, we initialise the weight $w_{u,i}$ and copy the current model $\Theta^{(t)}$ as a meta-model. The inner loop is a one-step optimisation of the $\nabla_w \mathcal{L}_{\mathrm{JOINT}}$ where $\mathcal{L}_{\mathrm{JOINT}}$ is evaluated using the meta-model. The value of weight is the gradient direction that maximises the $\mathcal{L}_{\mathrm{JOINT}}$. Then we rescale and normalize the weight vector to obtain the weight distribution that implicitly encodes the contribution of the data point towards diversity gain. Then the outloop updates the model parameter by the weighted MSE loss (which is relevance-only).

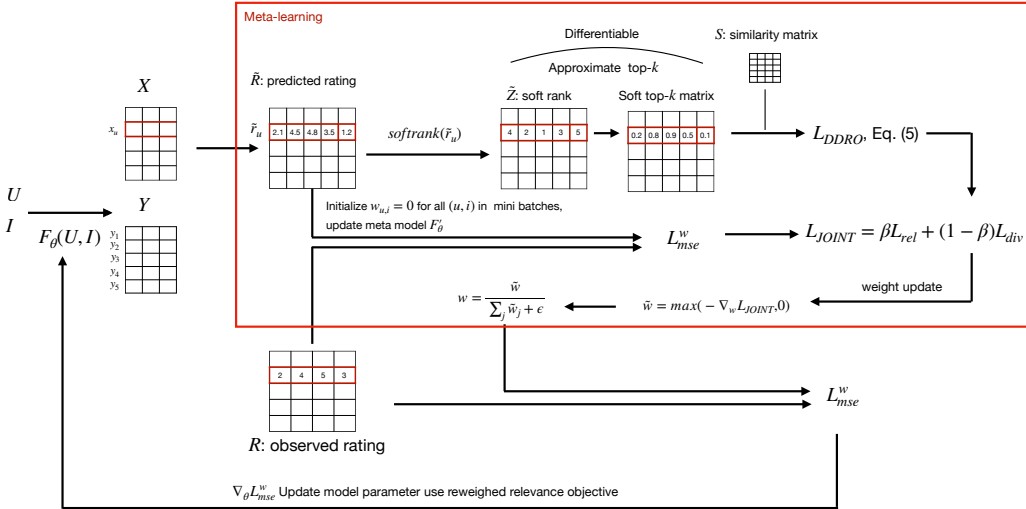

Figure 4: MDR workflow

**Direct diversity-guided tuning** To obtain top-$k$ relevant and diverse recommendations—without relying on post-hoc re-ranking or heuristic interventions—we propose an integrated approach. Our approach DDT integrates a $\beta$-balanced relevance-diversity loss into a standard end-to-end training pipeline. Specifically, we use gradient-based joint optimization, which promotes relevance and diversity. The overall procedure is outlined in Algorithm 1. This method preserves the simplicity and efficiency of conventional end-to-end training while explicitly aligning the optimization process with our dual objective. By embedding both goals within a single differentiable framework, DDT eliminates the need for separate re-ranking stages.

**Meta diversity-guided reweighting** Our second approach is inspired by meta-learning and bilevel optimization, where hyperparameters or per-example weights are optimized with respect to a validation objective (Ren et al., 2018). Here we ask *"Which training examples, if included in the update, will reduce both training loss and improve diversity?"* To this end, we estimate and assign importance weights $w_{u,i} \in [0, 1]$ to training samples $(u, i, r)$ for each batch $\mathcal{B} \subseteq \Omega_{\mathrm{T}}$, yielding

$$\mathcal{L}_{\mathrm{MSE}}^{w} = \sum_{(u,i)\in\mathcal{B}} w_{u,i} \left( \mathbf{R}_{u,i} - \tilde{\mathbf{R}}_{u,i} \right)^2 . \tag{10}$$

As shown in Algorithm 2, we begin each mini-batch update (line 4) by initializing the per-example weights $w_{u,i} = 0$, effectively ignoring all samples. We then perform a one-step inner update to obtain a temporary model $\Theta'$ using the weighted training loss $\mathcal{L}_{\mathrm{MSE}}^{w}$ (line 5–6), where $w$ is currently all zero. Next, we re-evaluate predictions using the updated parameters (line 7) and compute the joint meta-loss $\mathcal{L}_{\mathrm{JOINT}}$ (line 8), combining relevance and diversity. For this, we compute the gradient of $\mathcal{L}_{\mathrm{JOINT}}$ with respect to $w$, producing a utility score for each sample. We rectify it via $\tilde{\mathbf{w}} = \max(-\nabla_{\mathbf{w}}\mathcal{L}_{\mathrm{JOINT}}, 0)$ (line 9), and normalize (line 10) to obtain the final per-example weights $\mathbf{w}$, enforcing $\sum_{(u,i)\in\mathcal{B}} w_{u,i} = 1$. Normalization ensures that the overall gradient magnitude—and thus the effective learning rate—remains consistent across training steps, similar to standard Stochastic Gradient Descent (SGD), which averages over the batch (Ren et al., 2018). The final model update is then performed using this reweighted loss (line 11). This implicit diversity-guide optimization in MDR allows us to explore whether modifying the data sampling distribution alone can help achieve a decent relevance-diversity trade-off. We provide an empirical answer in Section 5, demonstrating that such implicit reweighting can be a practical and effective alternative to explicit multi-objective optimization.

---

**Algorithm 1** Direct Diversity-Guided Tuning (DDT)

---

**Require:** Initialization model parameter $\mathbf{\Theta}$, training data $\Omega_{\mathrm{T}}$, item distance matrix $\mathbf{D}$, trade-off $\beta$, learning rate $\eta$
1: **for** each epoch from 1 to $T$ **do**
2:    **for** each mini-batch $\mathcal{B} \in \Omega_{\mathrm{T}}$ **do**
3:       $\tilde{\mathbf{R}}_{u,i} \leftarrow \mathcal{F}_{\mathbf{\Theta}}(u,i)$, for all $(u,i,\mathbf{R}_{u,i}) \in \mathcal{B}$
4:       $\mathcal{L}_{\mathrm{MSE}} \leftarrow$ Compute $\mathcal{L}_{\mathrm{MSE}}(\tilde{\mathbf{R}}_{u,i}, \mathbf{R}_{u,i}; \mathcal{B})$ with (1)
5:       $\mathcal{L}_{\mathrm{DDRO}} \leftarrow$ compute DDRO with (5)
6:       $\mathcal{L}_{\mathrm{JOINT}} \leftarrow \beta \cdot \mathcal{L}_{\mathrm{MSE}} - (1 - \beta) \cdot \mathcal{L}_{\mathrm{DDRO}}$
7:       $\mathbf{\Theta} \leftarrow \mathbf{\Theta} - \eta \cdot \nabla_{\mathbf{\Theta}} \mathcal{L}_{\mathrm{JOINT}}$
8: **return** $\mathbf{\Theta}$

---

**Algorithm 2** Meta Diversity Reweighting (MDR)

---

**Require:** Initialization model parameter $\mathbf{\Theta}$, training data $\Omega_{\mathrm{T}}$, item distance matrix $\mathbf{D}$, trade-off $\beta$, learning rate $\eta$
1: **for** each epoch from 1 to $T$ **do**
2:    **for** mini-batches $\mathcal{B} \in \Omega_{\mathrm{T}}$ **do**
3:       $\tilde{\mathbf{R}}_{u,i} \leftarrow \mathcal{F}_{\mathbf{\Theta}}(u,i)$, for all $(u,i,\mathbf{R}_{u,i}) \in \mathcal{B}$
4:       $\mathbf{w} \leftarrow \vec{0}$
5:       $\mathcal{L}_{\mathrm{MSE}}^{w} \leftarrow \mathcal{L}_{\mathrm{MSE}}^{w}(\tilde{\mathbf{R}}_{u,i}, \mathbf{R}_{u,i}, \mathbf{w}; \mathcal{B})$ with (10)
6:       $\mathbf{\Theta}' \leftarrow \mathbf{\Theta} - \eta \cdot \nabla_{\mathbf{\Theta}} \mathcal{L}_{\mathrm{MSE}}^{w}$, update meta model.
7:       $\tilde{\mathbf{R}}_{u,i}' \leftarrow \mathcal{F}_{\mathbf{\Theta}'}(u,i)$, for all $(u,i,\mathbf{R}_{u,i}) \in \mathcal{B}$
8:       $\mathcal{L}_{\mathrm{JOINT}} \leftarrow \beta \cdot \mathcal{L}_{\mathrm{MSE}}^{w} - (1 - \beta) \cdot \mathcal{L}_{\mathrm{DDRO}}$
9:       $\tilde{\mathbf{w}} \leftarrow \max(-\nabla_{\mathbf{w}} \mathcal{L}_{\mathrm{JOINT}}, 0)$;
10:      $\mathbf{w} \leftarrow \frac{\tilde{\mathbf{w}}}{\sum_j \tilde{w} + \epsilon}$; normalization according to Ren et al. (2018)
11:      $\mathbf{\Theta} \leftarrow \mathbf{\Theta} - \eta \cdot \nabla_{\mathbf{\Theta}} \mathcal{L}_{\mathrm{MSE}}^{w}(\tilde{\mathbf{R}}_{u,i}, \mathbf{R}_{u,i}, \mathbf{w}; \mathcal{B})$
12: **return** $\mathbf{\Theta}$

---

## C    CONTROLLED TOY EXAMPLE ILLUSTRATING MDR

We present a controlled toy example to illustrate how Meta Diversity Reweighting (MDR) computes sample weights within a mini-batch and how the resulting weights favor diversity.

In high-level, we consider a mini-batch containing user–item interactions

$$\big\{(U_1, I_1), (U_1, I_2), (U_1, I_3)\big\},$$

with learnable nonnegative weights

$$\mathbf{w} = (w_1, w_2, w_3), \qquad \sum_j w_j = 1.$$

Intuitively, MDR assigns larger weights to items that contribute more to diversity. This is achieved by performing a one-step meta-update and computing the gradient of the joint relevance–diversity objective with respect to the weights. The resulting gradient encodes how increasing each sample's weight would influence diversity. After rectification and normalization, the weights form a probability distribution that implicitly captures each sample's contribution to diversity improvement.

We consider a single mini-batch from the first training epoch containing five interactions from the same user:

$$\begin{aligned}
\textbf{Users:} \quad & [5794, 5794, 5794, 5794, 5794], \\
\textbf{Items:} \quad & [1,\ 2,\ 3,\ 4,\ 5], \\
\textbf{Ratings:} \quad & [5.0,\ 5.0,\ 5.0,\ 5.0,\ 5.0].
\end{aligned}$$

We note that the items map to [534, 38, 1604, 1958, 2064] in the MovieLens dataset. All ratings are set equal to control for relevance effects, ensuring that weight differences arise primarily from the diversity objective.

We define the pairwise item distance matrix

$$\mathbf{D} = \begin{bmatrix} 0.0 & 1.0 & 0.5 & 1.0 & 0.0 \\ 1.0 & 0.0 & 1.0 & 0.0 & 1.0 \\ 0.5 & 1.0 & 0.0 & 1.0 & 0.5 \\ 1.0 & 0.0 & 1.0 & 0.0 & 1.0 \\ 0.0 & 1.0 & 0.5 & 1.0 & 0.0 \end{bmatrix}.$$

This matrix reveals the following structure:

- Cluster A: items 1 and 5 (distance 0).
- Cluster B: items 2 and 4 (distance 0).
- Item 3 lies between the clusters, with distance 0.5 to Cluster A and distance 1 to Cluster B.

Since diversity increases when selected items are far apart, we expect MDR to assign higher weight to representative items from different clusters, particularly item 3, which bridges both clusters.

Next, we give the step-by-step computation following Algorithm 2.

**Step 1: Weight initialization.** Weights are initialized as

$$\mathbf{w}^{(0)} = [0, 0, 0, 0, 0].$$

**Step 2: Meta-model update.** We copy the current model parameters $\Theta^t$ to obtain a meta-model

$$\Theta' \leftarrow \Theta^t,$$

and perform one gradient step on the weighted relevance loss:

$$\Theta' \leftarrow \Theta' - \eta \nabla_{\Theta'} \mathcal{L}_{\mathrm{MSE}}^{\mathbf{w}}(\Theta').$$

**Step 3: Compute weight gradient from joint objective.** We evaluate the joint relevance–diversity objective on the meta-model and compute

$$\nabla_{\mathbf{w}} \mathcal{L}_{\mathrm{JOINT}}(\Theta').$$

In this example, the gradient is

$$[-22449.561, \ 1675.825, \ -60342.375, \ -39423.586, \ -35152.387].$$

A negative gradient indicates that increasing the corresponding weight would reduce the joint loss (i.e., improve diversity and relevance).

**Step 4: Gradient rectification.** We convert the gradient into nonnegative importance scores:

$$\tilde{w}_j = \max\left(-\frac{\partial \mathcal{L}_{\mathrm{JOINT}}}{\partial w_j}, 0\right),$$

yielding

$$[22449.561, \ 0, \ 60342.375, \ 39423.586, \ 35152.387].$$

**Step 5: Normalization.** We normalize to obtain the weight distribution:

$$w_j = \frac{\tilde{w}_j}{\sum_k \tilde{w}_k + \epsilon},$$

resulting in

$$\mathbf{w} = [0.143, \ 0.000, \ 0.383, \ 0.251, \ 0.223].$$

As expected, the toy example clearly shows that the MDR improve diversity by rise the weight of items from diverse clusters: the MDR give the highest weight to "middle" item 3, and relatively high weight for one (item 4) from cluster A and one (item 5) from cluster B, while the ather item from same cluster is low or even 0. Interestingly, the weight of item 4 (B) is higher than item 5 (A), since item 3 has the highest weight (0.383) and it is closer to cluster A.

This controlled example demonstrates that MDR automatically assigns higher weights to samples that increase diversity, without explicitly encoding cluster information. The diversity signal emerges naturally from the gradient of the joint objective through the meta-learning step.

## D DETAILED EXPERIMENT SETTING.

**Datasets.** To evaluate our methods across different recommendation scenarios, we consider datasets from three domains: entertainment, product, and social recommendations, as detailed in Tab. 1. To cover the *entertainment* domain, we use Netflix (Bennett & Lanning, 2007) and MovieLens (Harper & Konstan, 2015) for user–movie recommendations, as well as Yahoo-R2 (Dror et al., 2012) for user–music recommendations. These datasets contain user ratings on a 5-point scale $[1, 5]$, along with genre or category annotations. For Netflix (respectively, Yahoo-R2), we randomly sample $3\,000$ items (respectively, $5\,000$) and retain users with at least 20 ratings (respectively, 100+ ratings). In the *product recommendation* setting, we use the Coat (Schnabel et al., 2016), which captures user-coat interactions in e-commerce. It contains $[1, 5]$ ratings and item 'meta' attributes. Finally, for the *social recommendation* scenario, we consider the KuaiRec (Gao et al., 2022), which is collected from a mobile video-sharing platform, which includes play duration, video length, and 'watch ratios' from $0$ (never watched) to $2$ (twice watched), which we linearly interpolate to 5-star ratings for consistency.

**Baselines.** We compare our algorithm against a broad set of state-of-the-art recommender-systems methods, as well as diversification techniques covering greedy, probabilistic, and graph-based strategies. To study the impact of diversification, we employ classical *Non-negative MF* (NMF) (Lee & Seung, 1999) as a relevance-only baseline that does not use any diversity mechanisms. We also include two baselines from the post-processing family: *Maximal Marginal Relevance* (MMR) (Carbonell & Goldstein, 1998) and *Diversity-weighted Utility Maximization* (DUM) (Ashkan et al., 2015), both greedy diversification techniques applied on top of NMF as the underlying model.

MMR greedily selects top-$k$ items that maximizes a weighted combination of relevance and dissimilarity with previously selected items. DUM, on the other hand, uses a submodular combination of relevance and category-based diversity reward. *Determinantal Point Processes* (DPP) (Chen et al., 2018) estimates the likelihood of item sets to be diverse and relevant as the determinant of an item-item similarity kernel matrix, from which we select the top-$k$ using a greedy selection. Finally, we include a recent embedding-based method, *Diversified GNN Recommender* (DGRec) (Yang et al., 2023), which introduces a diversity-aware aggregation mechanism into graph neural networks by selecting neighbors that maximize coverage over item categories.

**Evaluation criteria.** We evaluate model accuracy using metrics suitable for top-$k$ recommendations, *hit rate*, *precision*, and *recall* metrics. For each user $u$, let $\mathcal{R}_u$ denote the top-$k$ recommended items and $\mathcal{T}_u$ the set of ground-truth relevant items (i.e., rated above 4 in the test set). *Hit rate* measures whether at least one relevant item appears in $\mathcal{R}_u$; *precision* is the fraction of items in $\mathcal{R}_u$ that are in $\mathcal{T}_u$; and *recall* is the fraction of relevant items in $\mathcal{T}_u$ that are retrieved in $\mathcal{R}_u$.

However, the above requires known ground-truth. To evaluate the relevance of unseen items, we compute the potential user satisfaction as the *relevance score*

$$\text{Relevance}(u) = \frac{1}{k} \sum_{i \in \mathcal{R}_u} \mathbb{I}[p(\tilde{\mathbf{R}}_{u,i}) > \tau], \tag{11}$$

using preference likelihoods $p(\tilde{\mathbf{R}}_{u,i}) \in [0, 1]$.

**Experiment environment.** We conduct all our experiments on 2 AMD Epyc 7742 CPUs, 1 TB of RAM and 1 NVIDIA DGX-A100 GPU. Our code is written in Python v3.11.7. All results are averaged over 10 independent runs. We adopt the Adam optimizer during the optimization.

**Detailed Parameter setting.** In all experiment, we set $\epsilon = 10^{-4}$ for a exact rank approximation. During the optimization with adaptive $\beta$, we choose a learning rate $l = 0.01$ for the matrix factorization model, and $l = 0.001$ for the neural network model. We keep the Adam weight decay parameter as the default.

## E ADDITIONAL EXPERIMENT

We report the additional experiment results that are omitted in the main content. Figure 5 shows the adaptive $\beta$ based optimization of two datasets (MovieLens and Netflix) and two models (NMF and NCF). The patterns are consistently aligned with the analysis. Figure 6 shows the 4 accuracy metric

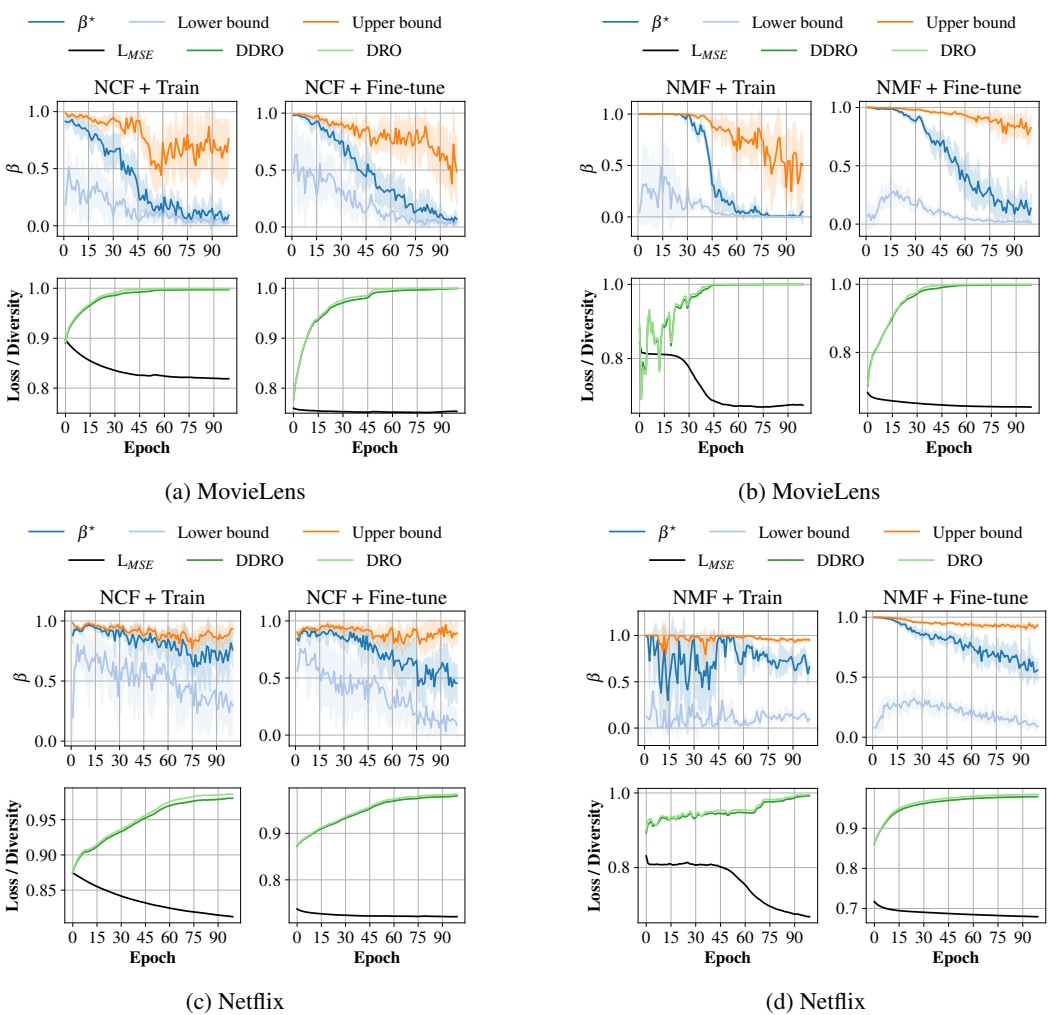

Figure 5: Optimization of relevance-diversity joint objective with adaptive $\beta$.

Table 5: Diversity gain achieved by Direct Diversity Tuning (DDT) on top-$1 \sim k$ and $k+1 \sim 2k$ recommendations across five datasets Coat, Yahoo-R2, Netflix, MovieLens, and KuaiRec. We vary the diversity reward parameter $k \in \{5, 10, 20, 30, 40\}$ and apply MDR to pre-trained NMF and NCF, reporting diversity gain. We report the mean and standard deviation across 10 runs.

| | Dataset | Diversity gain (%) | | | | | | | | | |
| | | $1 \sim k$ | | | | | $k+1 \sim 2k$ | | | | |
| | | $k=5$ | $k=10$ | $k=20$ | $k=30$ | $k=40$ | $k=5$ | $k=10$ | $k=20$ | $k=30$ | $k=40$ |
|---|---|---|---|---|---|---|---|---|---|---|---|
| NMF | KuaiRec | 8.5 (1.6) | 5.2 (2.0) | 6.5 (0.6) | 9.9 (1.3) | 8.7 (0.8) | 5.8 (1.3) | 10.1 (0.7) | 2.9 (0.9) | 10.3 (0.8) | 0.9 (0.8) |
| | Coat | 5.2 (3.4) | 2.7 (4.0) | 2.3 (1.1) | -1.2 (3.7) | 1.1 (1.3) | 0.4 (1.2) | 1.4 (0.9) | -0.4 (2.0) | 1.8 (0.7) | -1.2 (1.1) |
| | MovieLens | 36.4 (11.1) | 22.0 (9.9) | 32.3 (6.9) | 12.1 (4.4) | 25.4 (4.1) | 7.3 (2.9) | 21.7 (3.1) | 6.8 (1.7) | 19.7 (2.6) | 6.4 (1.4) |
| | Netflix | 14.2 (2.3) | 4.8 (3.2) | 13.0 (1.3) | 2.1 (2.1) | 10.3 (0.9) | 1.6 (1.7) | 9.2 (0.6) | 1.4 (2.0) | 8.7 (0.5) | 1.4 (1.5) |
| | Yahoo-R2 | 49.8 (19.1) | 36.0 (10.8) | 59.0 (12.4) | 34.2 (7.7) | 65.1 (5.8) | 33.1 (2.4) | 66.5 (3.2) | 31.4 (1.3) | 66.5 (2.3) | 29.6 (1.0) |
| NCF | KuaiRec | 1.3 (1.8) | -0.0 (1.2) | 0.3 (0.4) | 1.2 (1.8) | 1.1 (0.9) | 0.9 (1.2) | 1.3 (0.9) | 0.2 (0.9) | 1.4 (0.6) | 0.0 (0.8) |
| | Coat | 0.8 (2.3) | 0.9 (1.2) | 0.8 (0.9) | 0.3 (0.7) | 0.7 (0.9) | -0.3 (0.8) | 0.3 (0.6) | -0.1 (0.3) | 0.2 (0.4) | 0.1 (0.6) |
| | MovieLens | 33.2 (11.8) | 6.1 (4.3) | 17.8 (5.9) | 3.6 (2.3) | 9.5 (3.8) | 1.9 (2.0) | 4.9 (2.8) | 1.1 (1.3) | 3.4 (1.7) | 0.6 (1.0) |
| | Netflix | 4.6 (2.4) | 1.4 (1.4) | 2.8 (1.3) | -0.1 (1.1) | 1.1 (0.9) | -0.5 (0.8) | 0.4 (0.4) | -0.2 (0.3) | 0.2 (0.3) | -0.0 (0.3) |
| | Yahoo-R2 | 27.0 (15.3) | -0.2 (8.8) | 20.8 (10.2) | 2.3 (8.3) | 19.5 (8.5) | 7.3 (7.1) | 16.7 (8.1) | 2.0 (5.4) | 16.0 (6.9) | 3.2 (3.2) |

results: recall, mean squared error, hit rate, and precision, along with the diversity reword objective score (DRO). Similar to Table 3, Table 5 presents the *in-objective* and *out-objective* diversity gain of NMF and NCF with MDR.

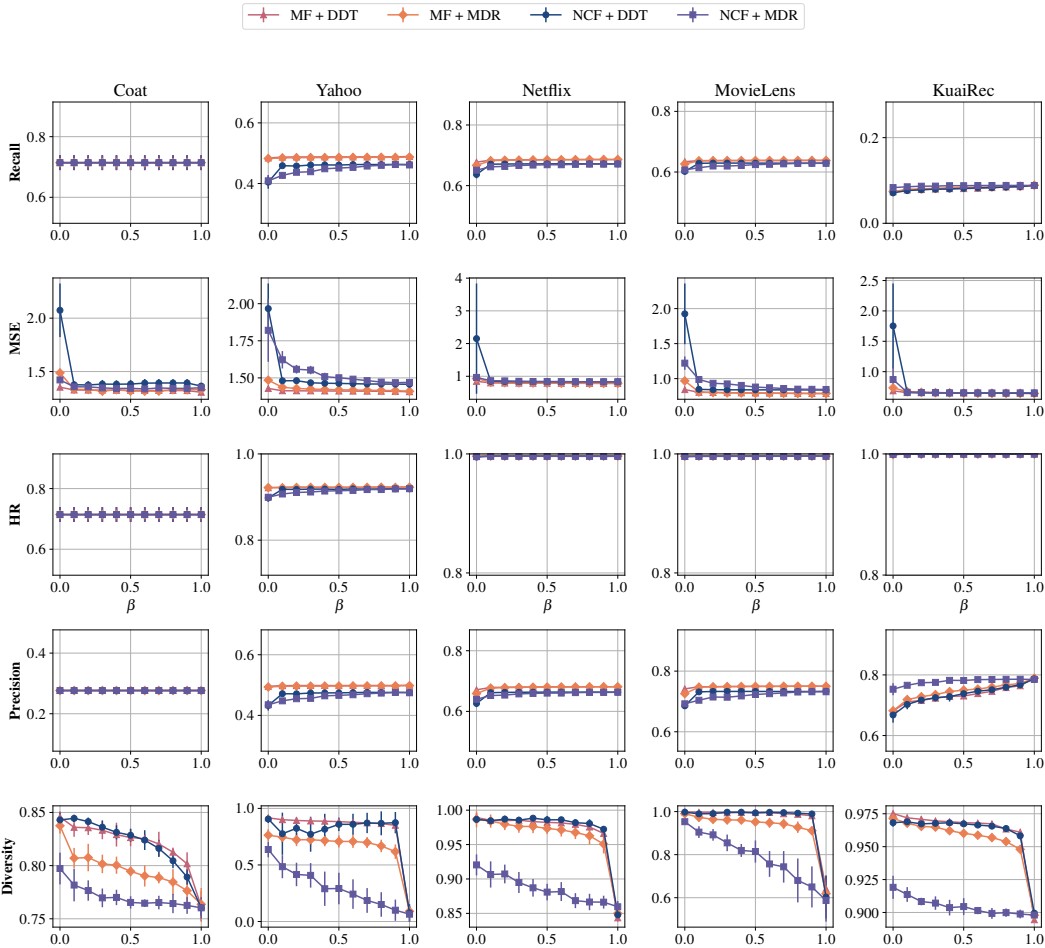

Figure 6: Performance comparison of DDT and MDR applied to two recommender models (NMF and NCF) across five datasets: Coat, Yahoo-R2, Netflix, MovieLens, and KuaiRec. We vary the parameter $\beta \in [0, 1]$ to control the trade-off between relevance and diversity, where $\beta = 1$ corresponds to optimizing only the relevance loss $\mathcal{L}_{\mathrm{MSE}}$ and $\beta = 0$ corresponds to optimizing only the diversity loss $\mathcal{L}_{\mathrm{DDRO}}$. The $x$-axis indicates the value of $\beta$. The $y$-axis shows recall (top row), mean squared error( second row), hit rate (third row), precision (fourth row), and diversity reword objective (DRO) score with $k = 10$ (bottom row). In each setting, we initialize from a pre-trained model (using $\mathcal{L}_{\mathrm{MSE}}$ only), then fine-tune with the joint loss $\mathcal{L}_{\mathrm{JOINT}}$ for 10 epochs, selecting the best result by diversity score. All experiments are repeated 10 times, and we report the mean and standard deviation.

| Metric | Before | DDT ($\beta = 0.2$) | DDT ($\beta = 0.5$) | DDT ($\beta = 0.8$) |
|---|---|---|---|---|
| BPR | 0.3125 | 0.3008 | 0.3008 | 0.2973 |
| Precision@10 | 0.0040 | 0.0040 | 0.0039 | 0.0040 |
| Recall@10 | 0.0396 | 0.0404 | 0.0394 | 0.0404 |
| HitRate@10 | 0.0396 | 0.0404 | 0.0394 | 0.0404 |
| NDCG@10 | 0.0191 | 0.0200 | 0.0200 | 0.0203 |
| MAP@10 | 0.0129 | 0.0139 | 0.0142 | 0.0143 |
| DDRO | 0.7776 | 0.8778 | 0.8813 | 0.8833 |
| DRO | 0.7803 | 0.8812 | 0.8815 | 0.8848 |

Table 6: Performance before and after DDT fine-tuning with BPR loss under different values of $\beta$. Each model is fine-tuned for 10 epochs.

