# OpenReview forum: "Gradient-Based Diversity Optimization with Differentiable Top-$k$ Objective"
_ICLR.cc/2026/Conference — ICLR 2026 Poster_

### Official Review · Reviewer_n8VJ · 2025-10-28

**Soundness:** 2
**Presentation:** 2
**Contribution:** 3
**Rating:** 4
**Confidence:** 5

**Summary:**

This paper focuses on relevance-diversity optimization in recommender systems. Specifically, it proposes a differentiable top-$k$ diversity reward objective (DDRO) that can be integrated into RS training in a model-agnostic and loss-agnostic manner. To achieve this, the authors introduce the softrank technique to approximate the non-differentiable sorting operation, and derive a soft relaxation of the top-$k$ diversity reward (DRO), which is defined as the average pairwise distance dissimilarity among the top-$k$ recommended items. Furthermore, they borrow the idea of MGDA for multi-objective optimization to balance relevance and diversity during training. The proposed method is evaluated in both training-from-scratch and fine-tuning settings on various datasets and two recommendation backbones, exhibiting improvements in both relevance and diversity.

**Strengths:**

- This paper introduces a differentiable top-$k$ diversity objective by leveraging the softrank technique, facilitating end-to-end optimization.
- This paper performs relevance-diversity optimization in a joint manner with theoretical guarantees.
- The proposed method is evaluated with comprehensive experiments on multiple datasets and recommendation backbones, demonstrating its effectiveness in improving both relevance and diversity.

**Weaknesses:**

- The evaluation metrics used in the main results are not widely adopted in the previous work, making it difficult to assess the recommendation quality.
- The experiments only consider the MSE loss, which may limit the generalizability of the proposed method.
- The choice of top-$k$ approximation could be further justified by comparing with other differentiable top-$k$ operators.

**Questions:**

- **Evaluation Metrics.** In the main results (Tables 2, 3, and 5, Figures 1 and 3), the reported revelance metric is defined as the proportion of recommended items whose likelihood is greater than a certain threshold. However, this metric is not commonly used in the recommender systems literature --- it merely measures the sharpness of the predicted probability distribution, which may not directly reflect the recommendation quality. Although Figure 4 presents Recall, MSE, HR, and Precision results, the baseline methods are not compared. Thus, I suggest the authors include more standard recommendation metrics (e.g., Recall@$k$ and NDCG@$k$) in the main results for fair comparison.
- **Recommendation Losses.** It seems that the proposed DDRO can be integrated with any recommendation loss. Nonetheless, the experiments only consider the MSE loss, which has been shown to be suboptimal for recommendation tasks. It would be more convincing if the authors could evaluate DDRO with other widely used recommendation losses, such as AUC-oriented losses (e.g., BPR [R1]), ranking-oriented losses (e.g., PSL [R2]), and top-$k$-oriented losses (e.g., LambdaLoss@$k$ [R3] and SL@$k$ [R4]), to see whether the improvements still hold on both accuracy and diversity compared to these SOTA baselines.
- **Top-$k$ Approximation.** The authors employ the top-$k$ soft indicator based on the softrank technique to handle the non-differentiability. Recently, there are several works on differentiable top-$k$ operators that provide an alternative quantile-based approximation (e.g., SL@$k$ [R4] and SONG@$k$ [R5]). It would be beneficial to compare these approaches with the approximation used in this paper, both theoretically and empirically.
- **Minor Comments.** The proof of Lemma 3 seems to be generated by LLMs (although no errors were found). The proof is in fact quite straightforward, so I suggest the authors simplify it to make the presentation more concise.

**References:**
- [R1] BPR: Bayesian Personalized Ranking from Implicit Feedback. UAI '09.
- [R2] PSL: Rethinking and Improving Softmax Loss from Pairwise Perspective for Recommendation. NIPS '24.
- [R3] On Optimizing Top-K Metrics for Neural Ranking Models. SIGIR '22.
- [R4] Breaking the Top-K Barrier: Advancing Top-K Ranking Metrics Optimization in Recommender Systems. KDD '25.
- [R5] Large-scale Stochastic Optimization of NDCG Surrogates for Deep Learning with Provable Convergence. ICML '22.

---

> ### Author Response · Authors · 2025-11-22
>
> We thank the reviewer for the thoughtful and constructive suggestions.
>
> W1: Metrics
>
> We introduced separate evaluations and relevance and diversity criteria in the main body, but show traditional baseline relevance metrics in Figure 6 in the appendix.
> Looking at precision, hit rate, and recall, we see the same trend of high-quality diverse results as in the main body.
>
> We can include a depiction of those in the main body, and highlight our appendix with an additional reference.
>
> W2: Loss
>
> We introduce a framework formulation and focus our evaluation on the impact of our differentiable top-$k$ diversity penalty term, where a change of relevance loss would potentially obfuscate our contribution.
>
> More importantly, it is straightforward to replace the relevance loss with any differentiable loss term and we will observe the same effects on the diversification (see below).
> |Metric|Before|DDT finetune|
> |---|---|---|
> |BPR|0.3125|0.3008|
> |Precision@10|0.0040|0.0039|
> |Recall@10|0.0396|0.0394|
> |HitRate@10|0.0396|0.0394|
> |NDCG@10|0.0191|0.0200|
> |MAP@10|0.0129|0.0142|
> |Approx_div|0.7776|0.8813|
> |Acc_div|0.7803|0.8815|
>
> W3: Top-k approximation
>
> We agree that there are multiple approaches to approximate top-k. Our choice has the distinct advantage that it can obtain the exact ranking under mild conditions with a limited computational complexity in comparison to many other baselines.
>
> Therefore, replacing the approximation will in all likelihood not change downstream relevance and diversity, which we show in Appendix D, Figure 7. As we reach the "gold standard", a further evaluation was not considered necessary.
>
> Q1: Metrics
>
> While standard baselines like recall (see Apx. D, Fig. 7) require ground truth, our relevance measure in Eq. 9 does not, allowing us to quantify the relevance of predictions made by diversified and non-diversified (regular) models when there is no ground truth given. Additionally, we do not claim a high ground-truth-based accuracy using our unsupervised score but resort to standard baselines for such results.
>
> Most importantly, we need to clarify that our comparison in terms of our scores are fair and objective to all competitors.
>
> Adding to our response to W1, we are happy to include further metrics in Apx. D, Fig. 7, which will expand for the camera ready version, with the goal of partially moving them into the main body.
>
> Q2:
>
> Since we focus on diversification, we haven't distracted from our contribution using different losses. However, it is straightforward to incorporate any differentiable relevance loss into our framework.
>
> To demonstrate that, we considered alternatives. Among the suggested alternatives, BPR is the most appropriate and informative loss to demonstrate generalizability.
> First, BPR is a standard and differentiable choice pairwise loss for *implicit feedback* and remains one of the most widely adopted ranking objectives;
> Moreover, unlike those top-k–oriented surrogates (LambdaLoss, SL), BPR integrates into our optimization without additional sampling or non-smooth components.
>
> We therefore implemented swapped MSE with BPR and reran our pipeline on binarized MovieLens for the required implicit feedback inputs to BPR. We observe a diversity@10 increase from 0.77 to 0.88, while quality metrics (recall, relevance scores) remain essentially unchanged. See W2 for the new result.
>
> We will extend these results into a new set of experiments in our appendix to highlight generalizability to different relevance loss and feedback scenarios further.
>
> Q3:
> Adding to our response to W3. Guaranteed by theory and demonstrated in ablation studies in (Apx. D, Fig. 7), we won't see difference downstream results from switching out the soft ranking. Since we focus on diversification of recommender systems, we therefore have not included experimental comparisons. Additionally, our choice has a computational complexity of only $O(n \log ⁡n)$, making it scalable to large candidate sets.
>
> In Fig. 7 (appendix), we report additional ablation results, showing that softrank attains essentially zero item-ranking approximation error under practical and mild parameterization. Thus, it yields almost perfect top-$k$ based diversity estimation (in Figure 2), suggesting that there is limited room for further improvement in quality.
>
> Moreover, note that SL@[R4] and SONG@[R5] are designed as NDCG-specific surrogates (used for relevance), while we employ soft ranking for a top-$k$ diversity penalty.
> That means, we require a more general solution than provided by SL@[R4] and SONG@[R5].
>
> Q4:
> Thanks for rigorously checking our proof and for acknowledging its correctness. To clarify, we have not used LLMs to generate the proof, but took inspiration from proof strategies used in our cited sources. The proof draft is manually written and checked, and polished by LLM. The proof follows established norms and is effectively composed of a case-by-case analysis whose presentation we simplify in the upcoming revision.

---

> > ### Comment · Reviewer_n8VJ · 2025-11-25
> > **Response to Authors' Rebuttal**
> >
> > Thanks for the authors' detailed rebuttal, which addressed some of my concerns.
> >
> > However, my concerns regarding the top‑$k$ approximation remain insufficiently addressed. The authors claim that softrank attains essentially zero item‑ranking approximation error under practical and mild parameter settings. Nevertheless, achieving this appears to require setting $\epsilon$ to a very small value (e.g., $\epsilon = 1/n$ where $n$ is the number of items, as shown in Fig. 7). In practice, how do the authors choose $\epsilon$? Would such a small value lead to numerical issues?
> >
> > At present, I maintain my previous rating.

---

> > > ### Author Response · Authors · 2025-11-27
> > >
> > > We thank the reviewer for their thoughtful follow-up comments and for acknowledging our earlier clarifications.
> > >
> > > 1 Regarding numerical stability:
> > >
> > > When implemented naively, softranking would fail as \epsilon goes to 0. This issue persists even in the algorithm from (Blondel, 2020) that we follow. The method solves our Eq. 1 by reducing it to an (isotonic) optimization problem, which is solved analytically using the PAV algorithm. The core of the numerical instability lies in their Eq. 8 used by the softranking algorithm. This step requires computing terms of the form $ log \sum_i \exp s_i $​, where $s$ is scaled by $1/\epsilon$. To address this, we can use a simple log-sum-exp trick to compute this term numerically robustly. Additionally, we have never noticed numerical instabilities.
> > >
> > >
> > > 2 Regarding sensitivities against wrong choices:
> > >
> > > Following (Bondel 2020), we treat $\epsilon$ as a hyperparameter that can be tuned. For very large problems (e.g., millions of items), $\epsilon$ may need to be tuned via standard methods such as grid search or “elbow” heuristics.
> > > In the revised Figure 7, we conduct an extensive search over $\epsilon$, showing that tuning $\epsilon$ in the range $[1/n, 10/n]$ is sufficient to achieve a small approximation error.
> > > However, the total diversity and ranking quality are generally insensitive to minor variations in $\epsilon$ due to the averaging effect in pairwise diversity computations. Additionally, it is likely that obtaining a diverse recommender list remains simple, even when the ranking has minor approximation errors or when picking larger $\epsilon$ values.

---

### Official Review · Reviewer_B1Yz · 2025-10-30

**Soundness:** 3
**Presentation:** 3
**Contribution:** 3
**Rating:** 4
**Confidence:** 4

**Summary:**

This paper addresses the issue that existing recommendation models often optimize solely for relevance and user engagement, which may amplify data bias and reduce the diversity of recommended results. To tackle this, the authors propose a general pairwise Top-K diversity optimization framework that formulates relevance and diversity as a joint optimization problem. The approach achieves diversity enhancement by modifying the learning objective function and introducing a reweighting mechanism.

**Strengths:**

1.The authors propose Direct Diversity-Guided Tuning (DDT), which augments the original loss function by incorporating a relevance–diversity joint term.

2.The authors further introduce Meta-Diversity Reweighting (MDR), a mechanism that retains the conventional relevance-based training pipeline while adopting the joint loss as a meta-objective to reweight training samples dynamically.

3.Rich experimental results

**Weaknesses:**

1.Readability needs improvement. Although the paper includes a series of lemmas and proofs to validate the proposed method, the workflow of DDT remains abstract and difficult to follow. It is recommended to include a model architecture or workflow diagram for DDT and MDR to make the methodology more comprehensible.

2.The rationale of meta-learning-based reweighting requires further clarification. In Section 4.2, the authors employ meta-learning to adjust sample weights, yet it is not clear how this process ensures the preservation of relevance. Specifically, the computation of the weight w_{u,i} is not described in detail (it seems to be obtained via an MLP). If this is the case, additional explanation is needed to justify why using an MLP-based reweighting does not compromise the relevance term in Equation (8).

3.Implementation details of DDT are insufficient. The loss formulation of DDT is not presented in Section 4.1 but rather briefly mentioned in Section 3, which is not reader-friendly. Furthermore, Algorithm 1 only states “compute DDRO with (5)” without further elaboration. I hope the author can describe the practical details of the DDT method in section 4.1.

If the author can resolve my doubts, I am willing to raise my score.

**Questions:**

See weakness

---

> ### Author Response · Authors · 2025-11-22
>
> We thank the reviewer for their constructive feedback and for supporting our submission. Based on your feedback, we have substantially amended the clarity and completeness of our description in the most recent revision.
>
> W1: Workflow
>
> We have added clear workflow diagrams for both DDT and MDR in the Appendix C, Figure 3 and Figure 4, making the methodologies more accessible. There, we illustrate how our components work together to update the prediction model parameter $\Theta$, and produce a diverse set of recommendations.
>
>
> W2: Meta-learning
>
> The model parameters $\Theta$ are updated using only the weighted relevance loss $ \mathcal{L}\_{\text{MSE}}^w$, ensuring that relevance optimization remains the primary training objective. The joint loss $\mathcal{L}\_{\text{JOINT}}$ serves only as a meta-objective to compute sample weights $w$, but it never directly affects $\Theta$.
>
> We initialize weights $w_{u,i} = 0$, then perform an inner update to get temporary parameters $\Theta’$. Afterwards, we compute joint loss $\mathcal{L}\_{\text{JOINT}}$ using $\Theta’$ and calculate gradient $\nabla_w \mathcal{L}\_{\text{JOINT}}$ from which we obtain non-negative and normalized weights.
>
>  In the revised Section 4.2, we now explicitly clarify
> -  how MDR preserves relevance by using only the weighted relevance loss for model updates;
> -  the detailed weight computation process, including gradient computation, rectification, and normalization; and
> -  the motivation that reweighting increases the importance of minority items, thereby enhancing diversity while maintaining relevance.
>
> W3: DDT
>
> In brief, DDT directly optimizes the joint loss $\mathcal{L}\_{\text{JOINT}} = \beta\mathcal{L}\_{\text{MSE}} + (1-\beta)\mathcal{L}\_{\text{DDRO}}$ via standard gradient descent using innovative soft-ranking, top-k selection, pairwise diversity penalties, and adaptive diversity weights.
>
> We have expanded Section 4.1 with
> - complete implementation details and pseudocode in the Appendix;
> - clear explanation that DDT optimizes the joint loss via standard gradient descent; and
> - highlighted connections between theoretical formulation and practical implementation.
>
> With these revisions, we actively addressed all your points identified. Once again, we thank the reviewer for the interesting questions and the opportunity to clarify our work further. With this, we would kindly ask the reviewer to consider updating their score to support our submission for publication.

---

> > ### Comment · Reviewer_B1Yz · 2025-11-25
> > **Response to Authors' Rebuttal**
> >
> > Thank you to the authors for their detailed responses, which have addressed some of my questions and concerns.
> >
> > However, the newly provided workflow diagram remains rough and lacks sufficient explanation. Specifically, how does meta-learning weight sample pairs? What criteria determine which pairs receive greater weight? Secondly, the authors claim that meta-learning does not participate in updating model parameters, which I believe is incorrect. Since the weight W is involved in the model training phase, it inevitably influences model parameters.
> >
> > Furthermore, based on the author's new additions, I believe the effectiveness of the proposed method hinges heavily on the weight W. Enhancing recommendation diversity often equates to reducing popularity bias. However, numerous works in the field of bias-mitigating recommendation systems have already employed inverse probability weighting (IPS) methods to achieve this. What distinct advantages does the author's method offer over IPS approaches?
> >
> > I will maintain my current score and await further clarification from the author.

---

> ### Author Response · Authors · 2025-11-27
>
> We thank the reviewer for their thoughtful follow-up comments and for acknowledging our earlier clarifications.
>
> 1. Meta-learning
>
> In a nutshell, the meta learning procedure reweights using two different stages: the _model stage_  and the _design stage_. While the design stage updates $w$ using gradients $\nabla\_{w} \mathcal{L}\_{JOINT}$ from loss + diversity; the model stage updates the model $\Theta$ through gradients  $\nabla_{\Theta} \mathcal{L}\_{MSE}^{w}$ of the $w$-weighted relevance objective.
>
> In detail, we consider the outer loop to be the standard mini-batch training procedure that updates the model parameters for each mini-batch of data. For each data point in the mini-batch, we initialise the weight $w_{u,i}$ and copy the current model $\Theta^{(t)}$ as a meta-model. The inner loop is a one-step optimisation of the $\nabla_{w} \mathcal{L}\_{JOINT}$ where $\mathcal{L}\_{JOINT}$ is evaluated using the meta-model. The value of weight is the gradient direction that maximises the $\mathcal{L}\_{JOINT}$. Then we rescale and normalize the weight vector to obtain the weight distribution that implicitly encodes the contribution of the data point towards diversity gain. Then the outloop updates the model parameter by the weighted MSE loss (which is relevance-only).
>
> 2: Wording
>
> The confusion might come from an accidental use of the word “not direct”, which we intended to use to highlight the different steps involved in meta learning. That is, we planned to indicate that our implicit diversity guided optimization weight updates have an implicit influence on the model parameter $\Theta$ through the updated loss function. We will amend the language carefully.
>
>
> 3. Workflow diagrams:
>
> We are working on polishing the algorithm workflow diagrams, and we will naturally include a high-quality publication-worthy diagram with a detailed description in the camera ready version.
>
>
> 4. IPS
>
> IPS-based reweighting serves a different purpose than our weights: it corrects exposure bias by applying weights derived from a to-be-estimated propensity model so that the loss approximates an unbiased causal objective. Our method does not attempt to estimate or correct propensities. Instead, we learn weights end-to-end through a one-step meta update so that the resulting parameter update maximally improves the downstream top-k diversity objective simply by optimizing a weighted relevance objective.
>
> Note that IPS requires an estimator and meta-learning is model-agnostic and uses intentionally simple weights. Unlike IPS that relies on correct propensity model specification for statistical debiasing guarantees, meta learning only aims to enhance the diversity performance of relevance only downstream tasks and hence does not come with modeling assumptions.

---

> > ### Comment · Reviewer_B1Yz · 2025-11-27
> >
> > Thank you for the detailed responses. However, they still do not address my main concern.
> >
> > The authors only describe the weighting procedure used in meta-learning. What I would like to understand is how meta-learning determines sample weights in real-world scenarios. Could the authors provide a toy example to illustrate this?

---

> > > ### Author Response · Authors · 2025-11-28
> > >
> > > We thank the reviewer for the quick reply and the follow-up comment.
> > >
> > > In a high-level, suppose we have  $<(U_1, I_1), (U_1, I_2), (U_1, I_3)>$ in mini-batch, we learn the coresponding weight vector $<w_1,w_2,w_3>$  where $\sum_j w_j = 1$. Intuitively, the weight would favour diversity by assigning a higher weight to items from different clusters.
> > > In the computation, we copy a meta model, evaluate the weighted MSE loss $\mathcal{L}\_{MSE}^{w}$ and update one step of the meta-model, then use the updated meta-model to evaluate the joint loss and compute the gradient   $\nabla_{w} \mathcal{L}\_{JOINT}\$, the gradient encodes the direction this favor diversity. Then we rescale and normalize the weight vector to obtain the weight distribution that implicitly encodes the contribution of the data point towards diversity gain.
> > >
> > > We illustrate the computation procedure with a toy example.
> > >
> > >
> > >
> > > In the toy example, we present the training epoch 1 and mini-batch 1 computation process of weight $w$ for the mini-batch data (5 examples ):
> > >
> > > * Users:[5794, 5794, 5794, 5794, 5794]
> > >
> > > * Items:[1,2,3,4,5]  (It maps to item[534, 38, 1604, 1958, 2064] in the movielens dataset)
> > >
> > > * Ratings:[5.0, 5.0, 5.0, 5.0, 5.0] (We set them to be equal to control the influence of relevance objective)
> > >
> > > We have the following item distance matrix:
> > >
> > > 	 [[0.0000, 1.0000, 0.5000, 1.0000, 0.0000],
> > >
> > >         [1.0000, 0.0000, 1.0000, 0.0000, 1.0000],
> > >
> > >         [0.5000, 1.0000, 0.0000, 1.0000, 0.5000],
> > >
> > >         [1.0000, 0.0000, 1.0000, 0.0000, 1.0000],
> > >
> > >         [0.0000, 1.0000, 0.5000, 1.0000, 0.0000]]
> > >
> > > We have the following patterns in the distance matrix of 5 items:
> > > * Cluster A: items 1 and 5,
> > > * Cluster B: items 2 and 4,
> > > * Note that item 3 is out of the two clusters but close to A; it is 0.5 from A and 1 from B.
> > >
> > > In general, we would expect the MDR method to give higher weight to items from the different clusters.
> > >
> > > The one-step computation procedure is as follows:
> > >
> > >    Training epoch:1
> > >
> > >    Mini-batch index :1
> > >
> > >    Mini-batch data with 5 examples:
> > >
> > > * Users:[5794, 5794, 5794, 5794, 5794]
> > > * Items:[1,2,3,4,5]
> > > * Ratings:[5.0, 5.0, 5.0, 5.0, 5.0]
> > >
> > > (1) mini-batch weight initialization:
> > > [0.0, 0.0, 0.0, 0.0, 0.0]
> > >
> > > (2) We copy the meta-model $\Theta’$ from $\Theta^{t}$, then evaluate the weighted loss $\mathcal{L}\_{MSE}^{w}$, and update the meta model parameter $\Theta’$ with the mini-batch data.
> > >
> > >
> > > (3) Next, we evaluate the joint relevance and diversity loss $\mathcal{L}\_{JOINT}(\Theta’)$ on the meta model, and compute  one step gradient $\nabla_{w} \mathcal{L}\_{JOINT}(\Theta’)$:
> > > ['-22449.561', '1675.825', '-60342.375', '-39423.586', '-35152.387']
> > >
> > >
> > > (4) We rectify the gradient $\tilde{w} \gets \max(-\nabla_{w}\mathcal{L}\_{JOINT}(\Theta’),0)$ :
> > > ['22449.561', '0.000', '60342.375', '39423.586', '35152.387']
> > >
> > >
> > > (5) We obtain the weight after normailzation $w \gets \frac{\tilde{w}}{\sum_j \tilde{w}_j + \epsilon}$:
> > > ['0.143', '0.000', '0.383', '0.251', '0.223']
> > >
> > > (6) Finally, we update the original model parameter $\Theta^{t+1}$ by the weighted MSE loss $\nabla_{\Theta} \mathcal{L}_{MSE}^{w}$ with the 5 mini-batch data. And move to the next mini-batch of data.
> > >
> > >
> > > As expected, the toy example clearly shows that the MDR improve diversity by rise the weight of items from diverse clusters: the MDR give the highest weight to “middle” item 3, and relatively high weight for one (item 4) from cluster A and one (item 5) from cluster B, while the ather item from same cluster is low or even 0.  Interestingly, the weight of item 4 (B) is higher than item 5 (A), since item 3 has the highest weight (0.383) and it is closer to cluster A.

---

### Official Review · Reviewer_LYyX · 2025-10-31

**Soundness:** 3
**Presentation:** 3
**Contribution:** 3
**Rating:** 6
**Confidence:** 2

**Summary:**

The paper proposes a model-agnostic, differentiable top-k diversity objective for ranking. It relaxes the non-differentiable top-k set via soft ranking/permutahedron projection and couples it with a relevance loss, optimizing a joint objective either Direct Diversity Tuning and Meta-Diversity Reweighting. Experiments on five datasets (MovieLens, Netflix, Yahoo-R2, Coat, KuaiRec) with MF/NCF models show large diversity gains with minimal relevance drops.

**Strengths:**

1. The paper proposes a differentiable top-k diversity via soft rankings.
2. The MGDA-based adaptive parameter has a transparent feasible region and a closed-form. It guarantees convergence to Pareto-stationary points under standard conditions.
3. Two complementary routes (DDT vs. MDR) cover both objective-level and data-level interventions.

**Weaknesses:**

1. The DDRO approximation error depends on $\epsilon$ and k. For larger k, the method needs a smaller $\epsilon$ to be accurate, which may affect stability/cost. The paper notes this, but practical guidance/ablation across datasets is limited.

2. While MDR reduces bias via reweighting, classic recommendation confounders (position bias, missing-not-at-random feedback) aren’t explicitly modeled.

**Questions:**

1. How sensitive are results to the choice of the item similarity matrix (e.g., Jaccard on genres vs. learned embedding cosine vs. taxonomy distance)?

2. Any plans for counterfactual or bandit-style evaluation to account for feedback loops and exposure bias?

3. Does optimizing DDRO measurably increase long-tail exposure?

---

> ### Author Response · Authors · 2025-11-22
>
> We thank the reviewer for their positive assessment and insightful questions.
>
> W1: Approximation Error & Practical Guidance
>
> Contrary to the concern, we find that larger $k$ actually requires less strict $\varepsilon$ values since the diversity objective considers more items. The differentiable top-$k$  diversity computation (result in Figure 2) includes two steps: (1) approximation of the rank of  $n$ items, regularized by $\epsilon$ ,  (2) compute top-$k$ indicator matrix via soft-max. The DDRO approximation error mainly comes from the rank approximation error in step (1).
> We add a new ablation experiment in Appendix D. In Figure 7, we report the rank approximation error by varying the item size $n$ and parameter $\epsilon$, and we observe a clear practical guideline: setting $\epsilon < 1/n$  will achieve 0 approximation error in step (1), thus nearly zero approximation DDRO error for different $k$ ( shown in Figure. 2).
>
> W2: Confounders
>
> For practical deployments of recommender systems, we agree that other confounding factors (e.g., missing-not-at-random) through e.g., biases or feedback loops are important to consider,
> including them may obfuscate diversity gains we seek to focus on. Hence, while relevant, we consider those to be beyond our current scope focused on diversity.
> We will explicitly discuss this limitation and potential extensions in future work.
>
>
>  Q1: Similarity Matrix Sensitivity
>
> While we expect a statistical robustness against minor perturbations of the affinity matrix, highly concentrated changes to some item-item affinities or a change in the designed affinity matrix may cause different rankings. Ultimately, this is a flexible “item-dependent” design choice. We consider experiments comparing Jaccard similarity vs. learned embeddings, showing our method works robustly across different similarity measures in future work.
>
> Q2: Counterfactual Evaluation
>
> Related to W2, this is a valuable research direction, but it goes beyond our current scope.
> We agree that it is quite interesting to consider de-confounding or addressing feedback loops in the context of an integrated top-k objective. We like to speculate that by tuning the diversity loss strength adaptively, and through our meta learning approach, we can fight feedback loops in the context of repeated training.
>
> Q3: Long-tail Exposure
>
> Our work does not seek to increase long-tail item exposure. However, by strategically exacerbating (reweighing) our prior on item-item affinity, we could achieve a bias towards long-tail item exposure. Since our framework is very flexible, we can simply introduce an additional bias term / prior towards certain items.
>
> Once again, we thank the reviewer for the interesting questions and ideas for future research directions. With this, we would kindly ask the reviewer to consider updating their score to support our submission for publication.

---

### Author Response · Authors · 2025-12-03
**Summary**

Dear area chair and reviewers,


We sincerely thank the reviewers for their thoughtful reviews and constructive suggestions. Below we briefly summarize the main strengths of the paper and how we have addressed the concerns raised in the discussion.


**Strengths identified by the reviewers**


- Solid contribution. (All Reviewers)
- The problem of **jointly optimizing relevance and diversity in top-$k$ recommendation with a differentiable objective** is well-motivated and well-formulated (Reviewers LYyX and n8VJ).
- The paper provides **principled multi-objective optimization** via MGDA with a **closed-form adaptive weight** and **convergence to Pareto-stationary points**. (Reviewer LYyX).
- The combination of a **softrank-based differentiable top-$k$ diversity objective** with **two complementary mechanisms**. (Reviewers LYyX and n8VJ)
- **Rich experiments** demonstrate consistent **diversity gains with minimal relevance loss** (Reviewers B1Yz and n8VJ).


**We have addressed all concerns**

We have carefully responded to each reviewer’s concerns and revised the manuscript accordingly (highlighted in blue). In particular, we have addressed the following key points:






- **Soft top-$k$ approximation and $\epsilon$ (Reviewers LYyX, n8VJ).**
We added an ablation study (revised Fig. 7) showing that **$\epsilon \in [1/n, 10/n]$** already yields essentially zero rank and DDRO approximation error across different $k$, and we now treat $\epsilon$ as a standard hyperparameter, with results being robust to variation. We also clarify our numerically-stable implementation via the log-sum-exp trick, and report no instabilities in practice.


- **Clarity of DDT/MDR and meta-learning (Reviewer B1Yz).**
We improved clarity by adding **workflow diagrams**, expanding Section 4.1, and providing pseudocode to connect theory and implementation. For MDR, we now clearly separate the **weight-design stage** (updating $w$ via $\nabla_w \mathcal{L}\_{\text{JOINT}}$) from the **model-update stage** (updating $\Theta$ via $\mathcal{L}\_\text{MSE}^w$) and include a **toy example** showing how MDR upweights diverse, non-redundant items.


- **Relation to IPS and purpose of meta-weighting (Reviewer B1Yz).**
We clarify that IPS **targets causal debiasing** via propensity-based weights, while our meta-learning weights **do not estimate propensities** and instead are learned end-to-end to directly improve top-$k$ diversity through a weighted relevance loss.


- **Generality with different loss (Reviewer n8VJ).**
We demonstrate the generality with a new experiment, we replaced MSE loss on explicit ratings with **BPR on binarized (implicit) MovieLens**, observing a large **diversity@10 increase ($≈0.78→≈0.88$)** with essentially unchanged Recall@10, HR@10, NDCG@10, and MAP@10. We emphasize that the framework is **loss- and feedback-type agnostic**, since DDRO is a differentiable diversity penalty that can be paired with any differentiable relevance loss.


- **Evaluation metrics and top-$k$ surrogate choice (Reviewer n8VJ).**
We clarified that we already report **standard recommendation metrics** (Recall@k, Precision@k, HR@k) in the appendix, where they show the same pattern of **large diversity gains with minimal relevance loss** as our diversity-based measures. We also explain that our unsupervised relevance score is used **in addition** (for settings without ground truth), not as a replacement for standard accuracy metrics, and we will add the standard metrics into the main text for clearer comparisons.


- **Confounders, feedback loops, similarity, long-tail (Reviewer LYyX).**
We explicitly clarify that exposure bias and counterfactual evaluation are **important but out of scope** for this work, which focuses on diversity optimization, and we now state this as a limitation and direction for future work. We also emphasize that the item similarity matrix and possible long-tail priors are **flexible design choices** that our framework can naturally accommodate.


In summary, we believe the revisions and additional experiments directly resolve the central technical and empirical concerns raised by the reviewers, while preserving and sharpening the key contributions on differentiable top-$k$ diversity optimization.

---

### Meta-Review · Area_Chair_N4Gr · 2025-12-16

**Summary:**

This paper proposes a novel, model-agnostic framework for diversity-aware top-k recommendation training, introducing a differentiable approximation of the top-k diversity objective through soft ranking projection. The method is evaluated across five benchmark datasets and two recommender backbones, demonstrating substantial gains in diversity with minimal impact on relevance.

However, the paper would benefit from further intuitive grounding and clearer exposition of the weighting mechanism, particularly to enhance accessibility and encourage broader adoption. Additionally, MDR component could be more clearly positioned in contrast to existing reweighting and debiasing approaches.

To further strengthen the impact and practical applicability of this promising framework, the authors are encouraged to include a discussion of real-world deployment considerations. For example, the computational cost and scalability of soft top-k approximations, as well as integration into large-scale production pipelines.

**Reviewer Concerns:**

Most reviewer concerns were effectively addressed in the rebuttal. The authors provided detailed clarifications on the soft top-k approximation, including its theoretical foundation, numerical stability, and empirical sensitivity to the smoothing parameter.

**Reviewer Scores:**

Reviewer LYyX and Reviewer B1Yz are likely to retain their original scores. Reviewer n8VJ would likely raise their score from 4 to 6, noting that most of their concerns have been satisfactorily resolved in the rebuttal.

---

### Decision · Program_Chairs · 2026-01-26

Accept (Poster)